# Predicted loss and gain of function mutations in *ACO1* are associated with erythropoiesis

Gudjon R. Oskarsson[1,2,9], Asmundur Oddsson [1,9], Magnus K. Magnusson[1,2,9], Ragnar P. Kristjansson[1], Gisli H. Halldorsson [1], Egil Ferkingstad [1], Florian Zink[1], Anna Helgadottir [1], Erna V. Ivarsdottir [1], Gudny A. Arnadottir [1], Brynjar O. Jensson[1], Hildigunnur Katrinardottir[1], Gardar Sveinbjornsson[1], Anna M. Kristinsdottir[1], Amy L. Lee[1], Jona Saemundsdottir[1], Lilja Stefansdottir[1], Jon K. Sigurdsson[1], Olafur B. Davidsson[1], Stefania Benonisdottir[1], Aslaug Jonasdottir[1], Adalbjorg Jonasdottir[1], Stefan Jonsson[1], Reynir L. Gudmundsson[1], Folkert W. Asselbergs [3,4,5], Vinicius Tragante[1,3], Bjarni Gunnarsson[1], Gisli Masson[1], Gudmar Thorleifsson[1], Thorunn Rafnar [1], Hilma Holm[1], Isleifur Olafsson[6], Pall T. Onundarson[2,7], Daniel F. Gudbjartsson [1,8], Gudmundur L. Norddahl[1], Unnur Thorsteinsdottir[1,2], Patrick Sulem [1✉] & Kari Stefansson [1,2✉]

Hemoglobin is the essential oxygen-carrying molecule in humans and is regulated by cellular iron and oxygen sensing mechanisms. To search for novel variants associated with hemoglobin concentration, we performed genome-wide association studies of hemoglobin concentration using a combined set of 684,122 individuals from Iceland and the UK. Notably, we found seven novel variants, six rare coding and one common, at the *ACO1* locus associating with either decreased or increased hemoglobin concentration. Of these variants, the missense Cys506Ser and the stop-gained Lys334Ter mutations are specific to eight and ten generation pedigrees, respectively, and have the two largest effects in the study (Effect$_{Cys506Ser}$ = −1.61 SD, CI$_{95}$ = [−1.98, −1.35]; Effect$_{Lys334Ter}$ = 0.63 SD, CI$_{95}$ = [0.36, 0.91]). We also find Cys506Ser to associate with increased risk of persistent anemia (OR = 17.1, P = 2 × 10$^{-14}$). The strong bidirectional effects seen in this study implicate *ACO1*, a known iron sensing molecule, as a major homeostatic regulator of hemoglobin concentration.

[1] deCODE genetics/Amgen Inc., Reykjavik, Iceland. [2] Faculty of Medicine, School of Health Sciences, University of Iceland, Reykjavik, Iceland. [3] Department of Cardiology, Division Heart & Lungs, University Medical Center Utrecht, Utrecht University, Utrecht, The Netherlands. [4] Institute of Cardiovascular Science, Faculty of Population Health Sciences, University College London, London, UK. [5] Health Data Research UK and Institute of Health Informatics, University College London, London, UK. [6] Department of Clinical Biochemistry, Landspitali, the National University Hospital of Iceland, Reykjavik, Iceland. [7] Department of Laboratory Hematology, Landspitali, the National University Hospital of Iceland, Reykjavik, Iceland. [8] School of Engineering and Natural Sciences, University of Iceland, Reykjavik, Iceland. [9] These authors contributed equally: Gudjon R. Oskarsson, Asmundur Oddsson, Magnus K. Magnusson. ✉email: patrick. sulem@decode.is; kstefans@decode.is

Hemoglobin is a globular protein tetramer in red blood cells and is the essential oxygen-carrying molecule in humans[1,2]. The oxygen-carrying role of hemoglobin is dependent upon the heme-iron group and red blood cells are sensitive to iron availability during red blood cell formation. Hemoglobin synthesis in red blood cell precursors is a process tightly regulated by several sensors. This includes the highly conserved cellular iron and oxygen sensing mechanisms that are linked through the cytokine erythropoietin (EPO), which stimulates precursor cells to differentiate into mature red blood cells[3]. Replacement therapy of recombinant human EPO has been used to treat anemia since the 1990s[4]. Abnormally low and high concentration of hemoglobin define anemia and polycythemia, which are a part of the pathology of several rare Mendelian disorders[5].

A large number of sequence variants have been associated with variation in hemoglobin concentration through genome-wide association studies (GWAS)[6,7]. In particular, a recent study using a combined cohort of the UK Biobank and interval studies revealed 140 sequence variants associated with hemoglobin concentration[8]. The majority of the variants reported were common and only 16 were low frequency (<5%) or rare (<1%). The majority of the 140 variants associated with hemoglobin concentrations are noncoding and it remains unclear which genes they affect. The largest reported absolute effect on hemoglobin of one of these variants was 3.1 g/L per allele, corresponding to 0.11 standard deviations (SD), for the intronic variant rs530159671 in *LUC7L*[8].

We have previously performed GWAS of many phenotypes using variants identified through whole-genome sequencing of a large fraction of the Icelandic population (up to 18%). These studies have uncovered associations of rare and low-frequency variants with numerous diseases and other traits[9–15]. Rare variants often represent recent mutations that can be traced to a single ancestor. Here, through GWAS meta-analysis of 684,122 individuals from Iceland and the UK[16], we focus on the rare missense and loss-of-function variants with large effects on hemoglobin concentration.

It is unusual to observe variants in the same gene that associate with a phenotype independently of each other. This is especially true when the observed variants are rare, coding, and have large opposing effects on a trait. Therefore, of the loci harboring common and rare variants associated with hemoglobin concentration, we chose to focus on the ACO1 locus to better understand the effects of sequence variation in this gene on erythropoiesis in humans. ACO1 is of particular interest as this is a well characterized gene in cell and animal models, but little has been reported on the effects of sequence variation on this gene in humans. We report eight variants associated with hemoglobin concentration in ACO1, encoding cytosolic aconitase 1 (aka iron-responsive element binding protein 1 (IRP1)), a protein involved in cellular iron sensing. These include six rare coding variants, where four associate with increased and two with decreased hemoglobin concentration.

## Results

In the meta-analysis we combined GWAS results on hemoglobin concentration from 286,622 Icelanders and 397,500 individuals from the UK (Supplementary Figs. 1 and 2, Supplementary Table 1). In Iceland, we tested 37.6 million sequence variants, identified through whole-genome sequencing of 28,075 Icelanders and subsequently imputed into 155,250 chip-typed individuals, as well as 285,664 of their first- and second-degree relatives (imputation info > 0.8 and MAF > 0.01%)[14]. Out of a total of 440,914 individuals with genotype information, 286,622 have

hemoglobin measurements available. In the UK, the GWAS was performed on 40 million markers (imputation info > 0.8), from the Haplotype Reference Consortium (HRC) reference panel, imputed into 397,500 chip-typed individuals of European ancestry from the UK Biobank[17] and hemoglobin measurements were available for all. In total, 43 million markers were tested in the meta-analysis. Associations were considered significant if the P value in the combined dataset was below a weighted, Bonferroni corrected, genome-wide significance threshold based on variant annotation[18] (significance thresholds in "Methods"). Heritability of hemoglobin concentration in the Icelandic population was estimated to be 0.20 (95% CI 0.19–0.21) and 0.29 (95% CI 0.29–0.30) using parent–offspring and sibling correlations, respectively (Supplementary Table 2).

We observe 334 loci harboring sequence variants reaching genome-wide significance (Supplementary Fig. 3 and Supplementary Data 1). We provide summary statistics for the GWAS meta-analysis of hemoglobin concentration in Iceland and the UK for all tested variants (Supplementary Data 1 and "Data availability" section). In total, 138 variants at 121 loci have previously been reported to associate with hemoglobin levels in populations of European descent, for which we provide robust replication (98%) and demonstrate consistency of effect in the Icelandic and UK datasets in the current study (Supplementary Data 2). We observe that genome-wide significant associations of 22 rare coding variants (MAF < 1%) were observed at 13 out of the 334 loci associated with hemoglobin level (Supplementary Fig. 3 and Supplementary Data 3). We observe independent rare coding variants with opposing effects at both the EGLN2 and ACO1 loci. Rare coding variants in EGLN2 were reported by Astle et al.[8], whereas none have been reported in ACO1.

Five variants in ACO1, encoding cytosolic aconitase 1, also known as IRP1, associate genome wide significantly with hemoglobin concentrations, of which three are coding and one common noncoding variant rs7045087 represents a previously reported intergenic association[8] (Table 1). Subsequently, we tested the 34 remaining coding variants in ACO1 for association with hemoglobin concentration and found three additional associations after accounting for multiple testing ($P < 0.05/34 = 1.5 \times 10^{-3}$) (Table 1 and Supplementary Data 4). In total, six of the eight variants in ACO1 are rare coding (MAF 0.01–0.48%) that independently associate with hemoglobin concentration with large effects (effect ranging from −1.61 to 0.63 SD) (Supplementary Figs. 4 and 5, Supplementary Table 3). Of these variants, three are only present in Iceland, one only in the UK, and two in both Iceland and the UK (Table 1 and Supplementary Data 5). We did not observe heterogeneity in the effects of these two variants between the two countries (Table 1, Supplementary Data 5). The two common variants at ACO1 are modestly correlated ($r^2 = 0.13$), but do not explain the effects of each other or the rare variants (Supplementary Table 3).

We tested the eight hemoglobin associated variants at the ACO1 locus for association with anemia and polycythemia (five phenotypes), seven blood cell indices, and five iron biomarkers (Supplementary Tables 4 and 5), resulting in a total of 136 (eight times 17) tests and we found 23 associations ($P$ value $< 0.05/136 = 3.7 \times 10^{-4}$) (Tables 2 and 3, Supplementary Data 6–8). All eight variants associating with hemoglobin also associate with red blood cell counts (RBC) and hematocrit (HCT) with similar significance, direction, and magnitude of effect, consistent with the high correlation between the three phenotypes. Hemoglobin concentration was used as the primary GWAS phenotype and the correlated phenotypes for lookup. None of the variants associate with mean corpuscular volume (MCV) and mean corpuscular hemoglobin concentration (MCHC) given the number of tests performed (Supplementary Data 6). Overall this indicates that

**Table 1 Variants in *ACO1* associating with hemoglobin concentration in the meta-analysis of the Icelandic and the UK datasets.**

| Position (Hg38) | rs name | Amin/Amaj | MAF Ice/UK (%) | Consequence | Iceland allele count | UK allele count | LD-class size | Effect in SD [95% CI] | P | P-het |
|---|---|---|---|---|---|---|---|---|---|---|
| chr9:32429450 | – | A/T | 0.02/– | Cys506Ser | 62 | – | 1 | –1.61 [–1.98, –1.35] | 3e–24 | – |
| chr9:32418355 | rs41305321 | T/C | 0.48/0.12 | Arg168Trp | 1,472 | 1090 | 1 | 0.22 [0.15, 0.27] | 4e–22 | 0.62 |
| chr9:32450189 | rs12985 | C/T | 35.9/37.1 | *78T>C | – | – | 4 | 0.03 [0.02, 0.04] | 4e–20 | 0.077 |
| chr9:32455264 | rs7045087[a] | C/T | 27.6/30.0 | Intergenic | – | – | 2 | –0.02 [–0.03, –0.01] | 3e–11 | 0.094 |
| chr9:32418475 | rs61753543 | G/A | 0.16/0.12 | Thr208Ala | 487 | 653 | 4 | –0.21 [–0.31, –0.11] | 3e–08 | 0.41 |
| chr9:32423348 | rs745558996 | T/A | 0.02/– | Lys334Ter | 67 | – | 7 | 0.63 [0.36, 0.91] | 6e–06 | – |
| chr9:32430494 | rs750337798 | T/A | 0.21/– | Asn549Ile | 616 | – | 2 | 0.20 [0.11, 0.29] | 7e–06 | – |
| chr9:32448929 | rs147876514 | T/C | –/0.01 | Arg802Cys | – | 65 | 1 | 0.43 [0.18, 0.68] | 9e–04 | – |

Effect is shown for the minor allele in standard deviations. Significance levels and effects are shown for the combined analysis. *Amin* minor allele, *Amaj* major allele, *MAF* minor allele frequency, *Consequence* consequence of sequence variants on transcript or protein level (NM_001278352.1 and NP_001265281.1) according to HGVS nomenclature, *LD* linkage disequilibrium, *LD-class size* total number of variants correlating with $r^2 > 0.8$ to the variant, *P-het* P value for test of heterogeneity between Iceland and the UK.
[a]Previously reported in Astle et al.[8].

**Table 2 Associations of variants in *ACO1* and other relevant hematological quantitative phenotypes in the Icelandic–UK meta-study.**

Quantitative traits

| HGVS/rs name | Hemoglobin (N = 541,187) | | RBC (N = 540,541) | | MCV (N = 541,101) | | WBC (N = 537,621) | | PLT (N = 540,716) | | Ferritin[a] (N = 100,054) | | IBC[a] (N = 57,971) | | Iron[a] (N = 73,572) | | Tf Sat[a] (N = 56,318) | |
|---|---|---|---|---|---|---|---|---|---|---|---|---|---|---|---|---|---|---|
| | Effect | P | Effect | P | Effect | P | Effect | P | Effect | P | Effect | P | Effect | P | Effect | P | Effect | P |
| p.Cys506Ser | –1.61 | 2.6E–24 | –1.682 | 7.5E–25 | 0.09 | 0.59 | –0.07 | 0.68 | 0.52 | 5.5E–03 | –0.77 | 4.1E–06 | 0.07 | 0.72 | –0.44 | 8.6E–03 | –0.31 | 0.10 |
| p.Arg168Trp | 0.22 | 3.7E–22 | 0.21 | 1.9E–20 | 0.003 | 0.90 | –0.02 | 0.52 | –0.03 | 0.29 | 0.001 | 0.95 | 0.03 | 0.50 | 0.07 | 6.1E–02 | 0.03 | 0.53 |
| rs12985 | 0.02 | 4.3E–20 | 0.021 | 1.1E–20 | –0.002 | 0.46 | 0.001 | 0.81 | 0.002 | 0.37 | 0.001 | 0.83 | 0.001 | 0.79 | 0.01 | 0.34 | 0.01 | 0.27 |
| rs7045087 | –0.02 | 3.3E–11 | –0.018 | 5.7E–14 | 0.006 | 9.0E–03 | –0.008 | 4.4E–04 | –0.003 | 0.16 | 0.001 | 0.65 | 0.01 | 0.38 | 0.001 | 0.67 | –0.01 | 0.16 |
| p.Thr208Ala | –0.18 | 2.6E–08 | –0.177 | 1.8E–08 | 0.03 | 0.30 | –0.02 | 0.48 | 0.03 | 0.42 | 0.04 | 0.51 | –0.22 | 2.2E–03 | 0.01 | 0.83 | 0.05 | 0.49 |
| p.Lys334Ter | 0.63 | 6.1E–06 | 0.624 | 1.4E–05 | 0.10 | 0.47 | –0.16 | 0.30 | –0.10 | 0.55 | 0.33 | 6.5E–02 | –0.39 | 9.9E–02 | –0.07 | 0.73 | 0.12 | 0.60 |
| p.Asn549Ile | 0.20 | 6.8E–06 | 0.225 | 7.2E–07 | –0.04 | 0.42 | –0.05 | 0.29 | –0.09 | 0.11 | –0.11 | 2.4E–02 | 0.03 | 0.71 | 0 | 0.94 | –0.04 | 0.52 |
| p.Arg802Cys | 0.43 | 9.1E–04 | 0.272 | 3.4E–02 | 0.009 | 0.94 | 0.007 | 0.94 | –0.20 | 0.13 | – | | – | | – | | – | |

N is the number of individuals measured for each parameter. Effect is shown in standard deviations for the minor allele. Significance levels and effects are shown for the combined analysis. HGVS is definition the mutation according to the Human Genome Variation Society nomenclature.
*MCV* mean corpuscular volume, *WBC* white blood cell count, *PLT* platelets, *IBC* iron binding capacity, *Tf sat* transferrin saturation.
[a]Parameters based only on the Icelandic dataset.

**Table 3 Associations of variants in *ACO1* and relevant hematological case-control phenotypes in the Icelandic–UK meta-study.**

| Case-control phenotypes | | | | |
|---|---|---|---|---|
| HGVS/ rs name | Persistent anemia (*N* cases = 21,072), (*N* controls = 690,293) | | Polycythemia (*N* cases = 38,624), *N* controls = 672,655) | |
| | OR | P | OR | P |
| p.Cys506Ser | 17.1 | 2.0E−14 | 0.01 | 1.0E−03 |
| rs12985 | 0.95 | 5.0E−07 | 1.06 | 9.0E−09 |
| p.Arg168Trp | 0.71 | 0.01 | 1.56 | 4.0E−09 |
| p.Arg802Cys | 0.29 | 0.12 | 1.31 | 0.71 |
| rs7045087 | 1.02 | 0.2 | 0.96 | 9.0E−05 |
| p.Thr208Ala | 1.15 | 0.3 | 0.59 | 4.0E−04 |
| p.Asn549Ile | 0.77 | 0.4 | 1.74 | 8.0E−06 |
| p.Lys334Ter | 1.4 | 0.7 | 3.44 | 4.0E−04 |

Persistent anemia is where an individual has all hemoglobin concentration measurements below defined threshold of anemia based on gender. The polycythemia phenotype was defined as individuals that were at least once measured to be above the defined threshold of polycythemia based on gender. Controls for both phenotypes were individuals never reaching the hemoglobin threshold level for definition of the phenotype based on gender. *N* cases is the number of individuals defined to have the phenotype based on hemoglobin measurements ("Methods"). *N* controls is the number of individuals that do not fulfill the criteria to be defined with the phenotype. Effect is shown in odds ratio for the minor allele. Significance levels and effects are shown for the combined analysis. HGVS is definition the mutation according to the Human Genome Variation Society nomenclature.
*OR* odds ratio.

*ACO1* sequence variants affect the number of red blood cells but not their size or the hemoglobin content of individual red blood cells. In Iceland, we detect an association of one of the variants, Cys506Ser, with increased serum ferritin levels but none of the other variants are significant after accounting for multiple testing (Table 2, Supplementary Table 6 and Supplementary Fig. 6, Supplementary Data 8).

*ACO1* is a cytosolic, RNA-binding protein that regulates the translation or stability of mRNAs encoding proteins for iron transport, storage, and use. ACO1 has an alternate function as cytosolic (c-) aconitase when an iron–sulfur ([4Fe–4S]) cluster is bound to it. The distribution of ACO1 between these mutually exclusive activities requires no new protein synthesis; iron excess or reduction promotes aconitase or RNA-binding activity, respectively. Assembly and disassembly of the [4Fe–4S] cluster appears to be an effective mechanism for regulating ACO1 activity, dependent on facile interchange between the two functional conformations[19]. When iron is low (or NO high, $H_2O_2$ high), the iron-responsive element (IRE)-binding activity of ACO1 increases and it binds IRE in the 5′ and 3′ untranslated region of mRNAs of many genes involved in iron regulation. When the concentration of iron increases, an [4Fe–4S] cluster binds to ACO1 to yield a functional aconitase, which interconverts citrate and isocitrate in the cytosol and becomes inactive as an IRE-binding protein due to a large conformational change[20]. Studies with ACO1 null mice[21,22] have shown that ACO1 regulates the hypoxia inducible factor 2alfa (HIF2α) in similar manner. HIF2α is a transcription factor that is a key regulator of many genes involved in erythropoiesis including EPO.

**Predicted RNA-binding gain-of-function variants.** *ACO1* variants that associate with decreased hemoglobin concentration are predicted to shift the balance of ACO1 function to increased RNA binding. The *ACO1* variant with the largest effect on hemoglobin, NP_001265281.1:p.Cys506Ser (chr9:32429450[A]), associates with decreased hemoglobin concentration (Effect = −1.61 SD,

corresponding to 24.6 g/L, $P = 2.6 \times 10^{-24}$, $MAF_{Iceland} = 0.019\%$) (Table 1, Supplementary Data 6). We observed no difference in effect on hemoglobin concentration between male and female carriers of Cys506Ser ($P = 0.59$, $n_{males} = 20$, $n_{females} = 26$) (Supplementary Fig. 7a). In Iceland, one in 2600 individuals are heterozygous for Cys506Ser (Supplementary Table 7), while it is absent from other sequenced populations such as the gnomAD database of 130,000 individuals. We observed 62 heterozygous carriers of Cys506Ser out of the 155k chip-typed Icelanders, all of whom belong to a single eight generation pedigree originating from ancestors born around 1780 in the South-Thingeyjarsysla county (Fig. 1). Consistent with the large effect on hemoglobin concentration, we detect an association of Cys506Ser with a high risk of persistent anemia (all hemoglobin measurements < 118 g/L for women and <134 g/L for men) (Table 3). Persistent anemia was observed in 15 (28.3%) of the 53 Cys506Ser carriers with hemoglobin measurements but only in 1.7% of the general population ($OR = 17.1$, $P = 2.0 \times 10^{-14}$). We do not observe significant association with other diseases in the Icelandic population, given the number of phenotypes tested (significance threshold: $P < 0.05/413 = 1.2 \times 10^{-4}$) (Supplementary Data 9). Cys506Ser associates with decreased RBC (Effect = −1.68 SD, $P = 7.5 \times 10^{-25}$) (Supplementary Data 6), but has no effect on MCV and MCHC, phenotypically consistent with predisposition to normocytic anemia (Table 2, Supplementary Data 6). Among the Cys506Ser carriers, we observe a lower ferritin concentration (Effect = −0.77 SD, $P = 4.1 \times 10^{-6}$, $N = 100,051$), but no association with other iron parameters (Table 2, Supplementary Data 8). Furthermore, no effect was seen on other hematopoietic lineages (platelets and white blood cells) (Table 2, Supplementary Data 6).

The Cys506Ser missense variant is at a highly conserved genomic location among mammalian species (GERP = 5.05, top 20% of the exome and 0.7% of the genome, Supplementary Table 8) and is one of three cysteine residues (Cys437, Cys503, and Cys506) involved in direct binding of the [4Fe–4S] cluster to ACO1[23–25] (Fig. 2). In vitro and mice studies have shown that transgenic expression of the Cys506Ser mutation abolishes the binding of the [4Fe–4S] cluster to ACO1, leading to a constitutively active RNA-binding state of ACO1, independent of iron concentration. Consistent with our observations in humans, the Cys506Ser mice develop anemia[26]. Furthermore, the association of Cys506Ser with lower ferritin levels in our data suggests translational inhibition through the 5′ IRE in ferritin. Taken together, these data show that Cys506Ser is a gain-of-function mutation that generates an ACO1 protein with constitutively active RNA-binding function leading to normocytic anemia in humans.

Another rare missense variant, NP_001265281.1:p.Thr208Ala, associates with decreased hemoglobin concentration (rs61753543 [G]) (Effect = −0.18 SD, $P = 2.6 \times 10^{-8}$) (Table 1). The variant has similar allele frequencies in Iceland and the UK ($MAF_{Iceland} = 0.16\%$, $MAF_{UK} = 0.12\%$, $P_{het} = 0.4$) (Supplementary Data 5). Thr208Ala is at a highly conserved genomic location among mammalian species (GERP = 5.79, top 4% of the exome and 0.2% of the genome, Supplementary Table 8) but does not fall within a known RNA binding or [4Fe–4S] cluster sites (Fig. 2). The association with decreased hemoglobin concentration suggests that Thr208Ala increases RNA-binding function of ACO1, either through increased RNA affinity (binding to IRE) or decreased binding of [4Fe–4S] cluster to ACO1.

**RNA-binding loss-of-function variants.** Variants that associate with increased hemoglobin concentration are predicted to decrease the RNA-binding function of ACO1. The stop-gained

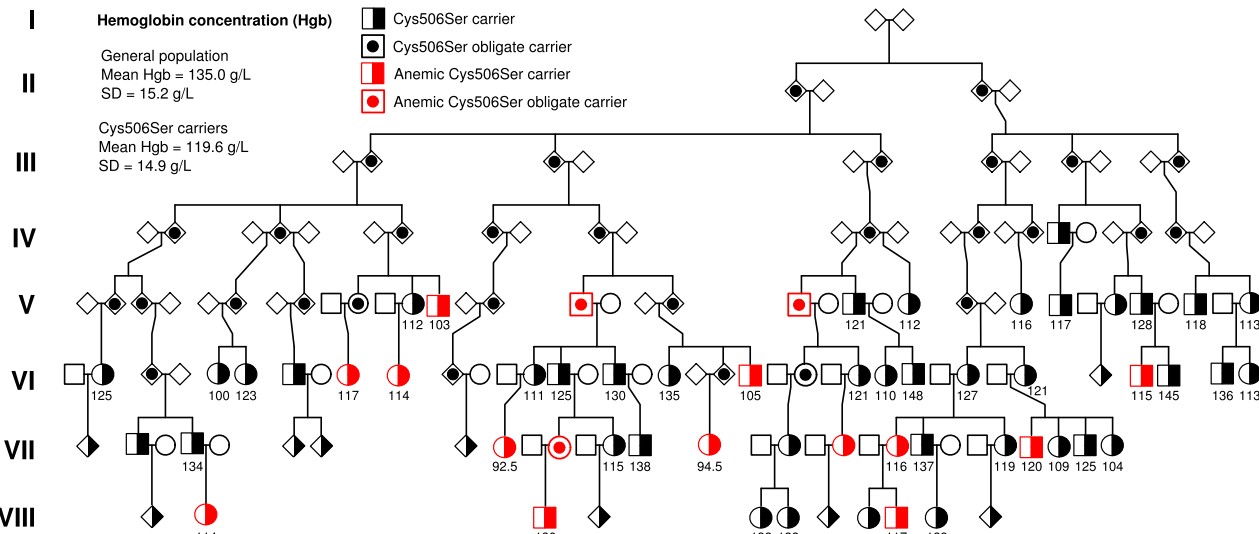

**Fig. 1 Pedigree of carriers of Cys506Ser in ACO1.** All 62 carriers can be traced back to ancestors born in the late 18th century. The founding couple had eight offspring and a current total number of 5,430 descendants. Roman numerals indicate generation, mean hemoglobin concentration is noted below the symbols. square = male, circle = female, diamond = sex unspecified, solid filled object = carrier, half filled object = obligate carrier, red filled object = persistent anemia.

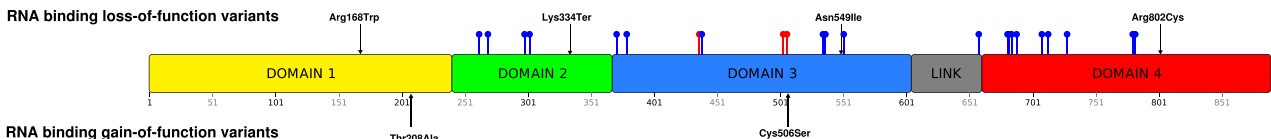

**Fig. 2 Schematic diagram of the ACO1 protein domain structure.** Central core domains 1 (yellow) and 2 (green), domain 3 (blue), linker (gray), and domain 4 (red) (based on Walden et al.[25]). Red lollipops represent cysteine residues required for iron–sulfur binding and aconitase function. Blue lollipops represent the amino acids required for binding to the iron-response element found on mRNA transcripts of various proteins required for stable erythropoiesis. Black arrows represent the sex rare coding variants we report associating with hemoglobin concentration. The axis is the numbers of each of the total 889 amino acids of ACO1. Protein reference: NP_001265281.1.

variant NP_001265281.1:p.Lys334Ter (rs745558996[T]), which is only detected in the Icelandic dataset, has the largest positive effect on hemoglobin concentration among *ACO1* variants (Effect = 0.63 SD, corresponding to 9.7 g/L, $P = 6.1 \times 10^{-6}$, $MAF_{Ice} = 0.023\%$) (Table 1, Supplementary Fig. 7b). Through its hemoglobin increasing effect Lys334Ter associates with increased risk of polycythemia (OR = 3.3, $P = 4.0 \times 10^{-4}$) (Table 3). No effects were seen on biomarkers for iron homeostasis (Table 2, Supplementary Data 8). However, the association of Lys334Ter with ferritin (Effect = 0.33 SD, $P = 0.065$) is consistent with the effect on hemoglobin concentration (Supplementary Table 6 and Supplementary Fig. 6).

In Iceland, 1 in 2200 individuals are heterozygous for Lys334Ter, while it is essentially absent from other populations, only observed in a single Finn in the gnomAD database[27]. We observed 67 carriers of Lys334Ter among the 155k chip-typed Icelanders, all of whom are clustered into a single ten generation pedigree originating from ancestors born in North Isafjardarsysla county around year 1710 (Fig. 3). Lys334Ter is located in exon 10 at position 334 out of 889 amino acids in the full-length protein (Fig. 2)[25]. Sequencing of RNA isolated from heterozygous carriers of Lys334Ter (N = 11) and noncarriers (13,152) demonstrated that transcripts containing Lys334Ter were present in the heterozygotes and that the amount of total RNA was 17.6% less in heterozygotes (95% CI −26.7 to −7.5%, $P = 0.0011$) than in noncarriers (Supplementary Table 9, Supplementary Fig. 8). These data are consistent with partial nonsense-mediated decay of the mutated transcripts. The presence of the mutated transcript

furthermore indicates that a truncated protein is generated. The truncated protein lacks domains 3 and 4 and part of domain 2 (Fig. 2) and will thus have lost its IRE-RNA-binding ability[25]. Consistent with loss of ACO1 RNA-binding function by Lys334Ter, homozygous knockout mice have increased hemoglobin concentration and polycythemia (Tables 2 and 3)[20].

Three additional rare coding variants in *ACO1* associate with increased hemoglobin concentration. First, the missense variant NP_001265281.1:p.Arg168Trp (rs41305321[T]) (Effect = 0.22 SD, $P = 3.8 \times 10^{-22}$) (Table 1) that is detected both in Iceland and UK (MAF in Iceland = 0.48%, and MAF in the UK = 0.12%). Arg168Trp also associates with increased risk of polycythemia (OR = 1.56, $P = 4.0 \times 10^{-9}$, hemoglobin levels > 152 g/L for women and >171 g/L for men) (Table 3, Supplementary Data 7). We identified seven homozygous carriers of Arg168Trp in the Icelandic dataset. The genotypic effect on hemoglobin concentration in homozygous carriers of Arg168Trp is consistent with an additive model (Supplementary Fig. 9). The variant is located within domain 1 of ACO1 shown to be important for IRE-RNAs binding of ACO1, suggesting that the variant might lead to reduced RNA binding (Fig. 2)[25,28]. Second, the missense variant NP_001265281.1:p.Asn549Ile (rs750337798[T]) (Effect = 0.20 SD, $P = 6.9 \times 10^{-6}$) (Table 1) is only found in the Icelandic population (MAF = 0.21%) (Supplementary Data 5). The variant is at a highly conserved genomic position (GERP = 6.05, top 0.8% of the exome and top 0.1% of the genome, Supplementary Table 8) and the amino acid substitution is located very close to Arg536, Arg541, and Lys551, which are critical for IRE-RNA

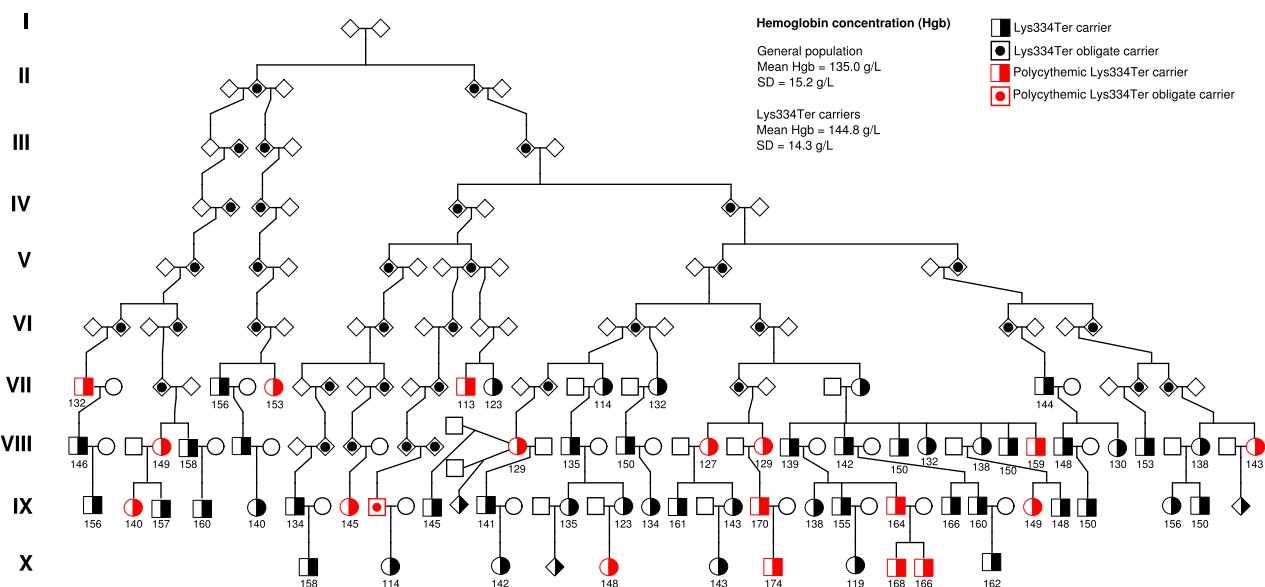

**Fig. 3 Pedigree of carriers of Lys334Ter in ACO1.** All 67 carriers can be traced back to a common ancestor in the early 18th century. The founding couple had six offspring and a current total number of 21,423 descendants. Roman numerals indicate generation, year of birth of the founding couple is noted above the symbols and mean hemoglobin concentration is noted below the symbols. Square = male, circle = female, diamond = sex unspecified, solid filled object = carrier, half filled object = obligate carrier, red filled object = polycythemic.

binding of the ACO1 protein[24,25] (Fig. 2). Third, a rare missense variant NP_001265281.1:p.Arg802Cys (rs147876514[T]) associates with increased hemoglobin concentration (Effect = 0.43 SD, $P = 9.1 \times 10^{-4}$) (Table 1). This variant is only detected in the UK (MAF = 0.01%) and is located within domain 4 of ACO1, which is important for IRE-RNA binding (Fig. 2).

**Common variants.** Two distinct common noncoding variants rs12985[C] and rs7045087[C] in *ACO1* associate with increased and decreased hemoglobin levels, respectively, ($r^2 = 0.13$) (Table 1, Supplementary Table 3). The intergenic variant rs7045087[C] has only one highly correlated variant (rs1133071 [G], $r^2 = 0.81$) and was reported by others to associate with a small effect with reduced hemoglobin levels, RBC and HCT[8] (Table 2). The other common variant, the 3′UTR variant rs12985 [C] associates with increased RBC and HCT. rs12985[C] has two highly correlated variants ($r^2 > 0.8$; rs10813817[C] intronic in *ACO1*, rs201050034[G] intronic in *DDX58*). As expected, rs12985 [C] associates with increased risk of polycythemia (OR = 1.06, $P = 9.0 \times 10^{-9}$) and decreased risk of anemia (OR = 0.97, $P = 4.0 \times 10^{-6}$) (Table 3). Neither rs12985[C] nor rs7045087[C] show a strong correlation ($r^2 > 0.8$) with the top cis-eQTL in the region and rs7029002[C] does not associate with hemoglobin concentration (Effect = −0.007 SD, $P = 0.076$) making it unlikely that the association of rs12985[C] and rs7045087[C] with hemoglobin is through an effect on expression.

**Previously reported hemoglobin associated variants.** We show association results for 175 reported associations of sequence variants with hemoglobin concentration, 138 of which have previously been reported in populations of European descent. The large majority of reported variants (N = 119) come from the hitherto largest hemoglobin GWAS reported by Astle et al.[8], where the UK biobank participated with 87k individuals[1], which comprises 22% of the UK biobank dataset used in the current study (Supplementary Data 2).

Out of the 138 variants reported in European populations, 131 were tested in both the Icelandic and UK datasets and all show a direction of effect that is consistent with the initial report. In

Iceland, 113 out of the 131 variants replicate (Supplementary Data 2 and Supplementary Fig. 10). For the combined Icelandic and UK datasets 129 out of 131 variants replicate. We also compared effects in standardized and raw scale (g/L) for the 131 hemoglobin associated variants reported in European populations to explore whether there is a difference in effect estimates between the Icelandic and UK datasets (Supplementary Data 2 and Supplementary Fig. 10). There are 27% higher effect estimates on the standardized scale in the UK dataset than in the Icelandic one (ratio of effect UK/Iceland = 1.27 (95% CI 1.23–1.32)). We note that the variance of raw hemoglobin concentration is higher in the Icelandic dataset than in the UK one (SD of raw hemoglobin concentration: Iceland = 15.5 g/L, UK = 12.2 g/L) (Supplementary Table 1). Once effect estimates are converted to raw scale (g/L) the effects are almost identical in the Icelandic and UK datasets (ratio of effect UK/Iceland = 1.02 (95% CI 0.99–1.06)) (Supplementary Fig. 10). Thus, it appears that the difference in effect estimates on the standardized scale between UK and Iceland can largely be explained by the higher variance in hemoglobin concentration in Iceland.

The UK and Iceland datasets included in the present analysis are diverse in regard to recruitment practices[9,29]. Despite differences in age, population coverage, number, and purpose of measurements between the Icelandic and UK datasets, which are reflected in differences in the distribution of raw hemoglobin values (Supplementary Table 1 and Supplementary Fig. 2), we still observe similar effect of sequence variants on hemoglobin concentration in the two datasets (Table 1, Supplementary Data 2, and Supplementary Fig. 10).

## Discussion

The aim of this study is to understand how sequence variants in *ACO1* affect hematopoiesis. After we identified a genome-wide significant association at the *ACO1* locus, we performed conditional analysis to search for secondary associations at the locus, focusing on variants with a predicted protein-coding effect. We describe two noncoding variants at the *ACO1* locus and six distinct rare coding variants in *ACO1* with minor alleles that associate with either decreased (Cys506Ser and Thr208Ala) or

increased (Lys334Ter, Arg168Trp, Asn549Ile, and Arg802Cys) hemoglobin concentration. These variants also associate with RBC and HCT where the direction and magnitude of effect is consistent with their association with hemoglobin. However, none of these variants associate with MCV and MCHC, indicating that *ACO1* sequence variants affect the production of RBC but not the hemoglobin content of each cell. Furthermore, the effects on hemoglobin range from −1.61 to 0.63 SD demonstrating that they affect the protein function both in the opposite manner and to a different degree.

The two variants in ACO1 with largest effects are both likely to have pronounced effects on protein function with the larger effect of Cys506Ser an order of magnitude larger than that of any previously reported sequence variants associating with decreased hemoglobin concentration: carriers have −1.61 SD less hemoglobin, which corresponds to 24.6 g/L. This leads to a very high risk of persistent anemia among carriers (OR = 17.1). Structural studies have shown that when the [4Fe–4S] cluster is intact, protein domain 4 is folded over and covers the [4Fe–4S] cluster within the central core formed by domains 1 and 2[30]. When the iron–sulfur cluster disassembles because of iron depletion (and/or because of oxidative degradation of the cluster) or when mutations in any of the [4Fe–4S] binding cysteines prevent cluster binding, domain 4 moves by a flexible hinge linker exposing the core domains. This allows the IRE structure to bind specifically to the protein[25,30,31]. The Cys506 residue is one of three cysteines directly involved in binding of the [4Fe–4S] cluster to ACO1. Mutating any of these three cysteines to serine leads to constitutive activation of the RNA-binding capacity of ACO1 as the [4Fe–4S] cluster is unable to bind ACO1[20,24,32]. The effect on serum ferritin levels we observe is direct evidence that the Cys506Ser is acting as a constitutively active variant. Constitutive RNA-binding activity presumably leads to anemia through binding of ACO1 to the IRE elements of one or more of IRE containing transcripts, involved in erythropoiesis, and affecting their translation. Two known IRE containing genes have direct links to erythropoiesis, EPAS1 (encoding HIF2α) and ALAS2, both carrying 5′UTR IRE elements[33].

The variant with the second largest effect on hemoglobin is the stop-gained variant Lys334Ter that associates with increased hemoglobin concentration and increased risk of polycythemia. The variant is within domain 2 of the protein and is thus predicted to truncate the protein at amino acid 334 out of the 889 amino acid of the full-length protein. The truncated protein lacks fraction of domain 2 together with domains 3 and 4 and thus predicted to have lost its RNA-binding capacity[30]. Furthermore, sequencing of RNA from the blood of heterozygous carriers of Lys334Ter showed a 17% reduction in total RNA compared with noncarriers. Although, disruption of ACO1 has not been linked to Mendelian condition in humans, our data are consistent with Aco1 homozygous knockout mice that show symptoms of polycythemia and pulmonary hypertension, suggested to be caused by translational derepression of HIF2α (EPAS1) and subsequent elevation of serum EPO levels from the kidney and endothelin-1 levels from pulmonary endothelial cells[21,22]. There are no reports of pulmonary hypertension in carriers of Lys334Ter although it should be emphasized that we found no homozygous carriers. We speculate that the coding variants associated with increased hemoglobin concentration likely reduce the IRE-binding activity of ACO1, though it is not clear how that would happen based on co-crystal structures. None of the four coding variants identified are in close proximity to the amino acids known to most adversely affect IRE binding: Arg269, Lys379, Ser371, and Ser681. However, Asn549Ile is close to Lys551, which binds A15 of the IRE.

Both variants Cys506Ser and Lys334Ter are pedigree-specific and are of independent origin. We traced both back to common ancestors born in the 18th century, eight and ten generations ago. It is of note that identification of the pedigrees was not based on a priori knowledge of membership but rather enabled by sampling a large fraction of the population. A few factors are necessary for the detection of associations with such recent variants. First, the whole-genome sequencing of a large fraction of the Icelandic population (~9%) allows the detection of these rare variants. Second, the large fraction of chip-typed Icelanders (~50% or 155k) enables imputation of these variants into a reasonable number of carriers. Third, hemoglobin concentration is available for vast majority (93%) of chip-typed Icelanders and a large fraction of their relatives.

Finding several variants in the same gene that affect the function of the protein it encodes can lead to a better understanding of the role of the protein in both normal and abnormal biology. Here we report sequence variants with both loss- or gain-of-function that affect the same gene. Loss-of-function variants allow the identification of processes for which a gene is required, while gain-of-function variants in the same gene indicate that the gene is able to control the process it affects in a rheostatic manner[34]. The effects the *ACO1* variants have on hemoglobin and ferritin, either increasing or decreasing levels, suggest a regulatory function of ACO1 with effects that go both ways. The effects of the loss-of-function variants reported here most likely result from ACO1 haploinsufficiency, as we demonstrate for the stop-gained heterozygotes for Lys334Ter. The underlying mechanism of gain-of-function variants are usually harder to explain[34]. In case the of the Cys506Ser variant, the mechanism is well studied in model systems and is the result of a gain of IRE binding. Other coding variants in *ACO1* that produce similar phenotypic effects are most likely to go through the same mechanism of action. Both loss- and gain-of-function variants in PCSK9 have been identified that decrease and increase cholesterol levels, respectively, and led to the development of PCSK9 inhibitors to reduce LDL cholesterol levels[35]. Also, loss- and gain-of-function variants in *SCN9A* encoding a voltage-gated sodium channel cause syndromes encompassing decrease and increase in pain perception, respectively, and have triggered efforts to develop SCN9A inhibitors as a therapeutic[36]. The identification of loss- and gain-of-function variants in *ACO1* sheds light on mechanisms that could be exploited in the development of therapies targeting erythropoiesis. We provide evidence for ACO1 as a potential drug target for treatment of disorders of erythropoiesis.

## Methods

**Study subjects**. The meta-analysis combined the results of two GWAS of hemoglobin concentration. The Icelandic dataset consisted of hemoglobin concentration measurements from 1993 to 2016 of 286,622 Icelanders available from four different laboratories in Iceland. In the UK Biobank, hemoglobin concentrations from 397,500 participants were measured from 2007 to 2010.

All participating Icelandic individuals who donated blood, or their guardians, provided written informed consent. The family history of participants donating blood was incorporated into the study by including the phenotypes of first- and second-degree relatives and integrating over their possible genotypes.

All sample identifiers were encrypted in accordance with the regulations of the Icelandic Data Protection Authority. Approval for the study was provided by the Icelandic National Bioethics Committee (ref: VSNb2015010033-03.12).

**Genotyping**. The Icelandic part of the study is based on testing variants identified by whole-genome sequence (WGS) data from 28,075 Icelanders participating in various disease projects at deCODE genetics, sequenced using Illumina standard TruSeq methodology to an average genome-wide coverage of 37×. The effects of sequence variants on protein-coding genes were annotated using the variant effect predictor using protein-coding transcripts from RefSeq. We carried out chip typing of 155,250 Icelanders (around 50% of the population) using Illumina SNP arrays as previously described[9,37]. The chip-typed individuals were long-range phased[38], and

the variants identified in the whole-genome sequencing of Icelanders were imputed into the chip-typed individuals (Imputation info > 0.8 and MAF > 0.01%). In addition, genotype probabilities for 285,644 untyped close relatives of chip-typed individuals were calculated based on Icelandic genealogy. The whole-genome sequenced samples were variants called jointly and the sequence variants found through whole-genome sequencing were phased jointly. The process used for WGS sequencing of Icelanders, and the subsequent imputation from which the data for this analysis were generated has been extensively described in recent publications[9,37].

Genotyping of UK biobank samples was performed using a custom-made Affymetrix chip, UK BiLEVE Axiom[39], and with the Affymetrix UK Biobank Axiom array[40]. Imputation was performed by Wellcome Trust Centre for Human Genetics using variants identified from 32,488 WGS individuals in the HRC and the UK10K haplotype resources[17]. This yields a total of 96 million imputed variants, however only 40 million variants were imputed into 408,658 participants using the HRC reference set due to quality issues with the remaining variants.

**Determination of sequence variant origin.** Close to complete genealogical records of the Icelandic population are available dating back to the Icelandic national census of 1703, and incomplete records dating back to the settlement of Iceland in 874 CE[41,42]. The Icelandic genealogy coupled with the large fraction of the population that has been chip-typed allows us to determine the origin of sequence variants through long-range phasing and haplotype imputation[9]. We used the Icelandic genealogy database[42,43] to identify the most recent common ancestors of carriers of the two rarest ACO1 sequence variants, Cys506Ser and Lys334Ter. In both cases, all carriers shared a common ancestor. These sequence variants are absent from descendants of close relatives of the common ancestor carrying the same haplotype background.

**Phenotypes.** Hemoglobin measurements: In the Icelandic part of the study, we used 4,354,272 hemoglobin concentration measurements from 286,622 Icelanders from four different laboratories in Iceland from 1993 to 2018 (Supplementary Tables 1, 4, and 5). Of the 286,622 individuals with hemoglobin measured, 143,682 were chip-typed and 142,940 were first- or second-degree relatives of chip-typed. The geometric mean for number of measurements per subject is 6.4. In the laboratories, hemoglobin concentration was measured using routine automated and semiautomated hematology analyzers. Hemoglobin concentration measurements for each sex and the four different laboratories were separately transformed to a standard normal distribution and adjusted for age using a generalized additive model[17,44].

From the UK Biobank we used 418,628 hemoglobin concentration measurements from 397,500 individuals of white British ancestry, whose samples were collected at the UK Biobank assessment centers (Field ID 30020, hemoglobin concentration) (Supplementary Table 10). The median for number of measurements per subject is one measurement. The samples were processed and analyzed at the centralized processing laboratory of UK Biocenter using clinical hematology analyzers. The hemoglobin concentration measurements were adjusted for age and sex and population stratification using 40 principal components.

Hemoglobin concentration measurements as well as other basic hematology parameters used in the expression correlation were measured on EDTA anticoagulated blood using the Sysmex XN-1000 hematology analyzer.

Anemia: When deriving anemia status from the hemoglobin concentration measurements, an individual was defined to be anemic if all of their measured hemoglobin concentrations were below the anemic diagnostic threshold (below 118 g/L for women and below 134 g/L for men).

Heritability: Heritability of hemoglobin concentration was estimated in the following two ways: (1) 2 × parent–offspring correlation, (2) 2 × full sibling correlation, using the Icelandic data (where all family relationships are known).

**Association analysis.** We performed a meta-analysis on 286,622 individuals from Iceland and 397,500 individuals from the UK Biobank with at least one hemoglobin concentration measurement. In Iceland, quantitative traits were tested using a linear mixed model implemented in BOLT-LMM[45]. We tested 37,592,353 variants (with imputation info > 0.8 and MAF > 0.01%) identified from the whole-genome sequencing of 28,075 Icelanders (~9% of the population) for association with hemoglobin concentration. For binary phenotypes, sex, county of birth, current age or age at death (first- and second-order terms included), blood sample availability for the individual, and an indicator function for the overlap of the lifetime of the individual with the time span of phenotype collection were included as covariates in the logistic regression model. In the UK Biobank study, 40 principal components were used to adjust for population stratification and age and sex were included as covariates in the logistic regression model and the BOLT-LMM. The quantitative traits were transformed to a standard normal distribution. Only white British individuals were included in the study.

For the meta-analysis we used a fixed-effects inverse variance method based on effect estimates and standard errors from the Icelandic and the UK Biobank study[46]. For each study we used linkage disequilibrium (LD) score regression to account for distribution inflation in the dataset due to cryptic relatedness and

population stratification[47]. Using a set of about 1.1 million sequence variants with available LD score, we regressed the $\chi^2$ statistics from our GWAS scan against LD score and used the intercept as correction factor. The estimated correction factor for hemoglobin concentration based on LD score regression was 0.68 for the additive model in the Icelandic sample and 1.40 in the UK Biobank. In Iceland, when testing the association of sequence variants with quantitative traits, a BOLT linear mixed model was applied. These models are now widely used as they account for cryptic relatedness while also increasing power[45]. One-step in the BOLT-LMM procedure (step 1b) is to calibrate the $\chi^2$ test statistic by calculating a constant calibration factor. To compute the calibration constant BOLT-LMM rapidly computes the prospective statistic at 30 random SNPs by applying conjugate gradient iteration. However, this scaling was not applied to the test statistic in the Icelandic association model. Therefore, when we applied the LD score regression and estimate a correction factor from the regressions intercept it was shifted by this constant factor. The correction factor can thus indeed be below one due to the calibration factor (Supplementary Fig. 11). The intercept is therefore not comparable with correction factors obtained from standard genomic control methods, and should not be interpreted as such. Expected allele counts for sequence variants were used as covariates in the regression to test for association with other sequence variants conditional on their effects.

**Significance thresholds.** We applied genome-wide significance thresholds corrected for multiple testing using a weighted Bonferroni correction that controls the family-wise error rate. Based on variant annotation classes the weights used are the predicted functional impact of the class[18]. A total of 45,078,764 sequence variants were tested in either deCODE or the UK Biobank data. The adjusted significance thresholds are $1.8 \times 10^{-7}$ for variants with high impact ($N = 12,456$), $3.5 \times 10^{-8}$ for variants with moderate impact ($N = 235,454$), $3.2 \times 10^{-9}$ for low-impact variants ($N = 3,334,594$), $1.6 \times 10^{-9}$ for other variants in Dnase I hypersensitivity sites ($N = 5,928,505$), and $5.3 \times 10^{-10}$ for all other variants ($N = 35,567,755$).

Given that we observe genome-wide significant associations to hemoglobin levels corresponding to coding variants in ACO1, we decided to test all ACO1 coding variants with hemoglobin levels. In total, we tested 34 coding variants in ACO1 and apply a Bonferroni correction significance threshold of $0.05/34 = 1.5 \times 10^{-3}$.

**Sanger sequencing and re-imputation.** Sequence variant T>A at chr9:32,429,450 (hg38), corresponding to p.Cys506Ser, was poorly imputed due to the low frequency of the variant and low number of sequenced carriers in the original Icelandic dataset. A group of probable carriers of p.Cys506Ser and noncarriers were Sanger sequenced, and re-imputation was subsequently carried out in the same population. Sanger sequencing confirmed 58 carriers of Cys506Ser and 62 were identified after re-imputation. Imputation information following re-imputation increased from 0.94 to 0.97.

**RNA-sequencing analysis.** RNA sequencing data from whole blood of 13,174 individuals from Icelandic samples. Gene expression was computed based on personalized transcript abundances[48]. Association between variant and gene expression was estimated using a generalized linear regression assuming, additive genetic effect and normal quantile-transformed gene expression estimates, adjusting for measurements of sequencing artefacts, demography variables, blood composition, and hidden covariates[49]. All variants within 5 Mb of each gene were tested.

**Reporting summary.** Further information on research design is available in the Nature Research Reporting Summary linked to this article.

## Data availability

Sequence variants passing GATK filters have been deposited in the European Variation Archive, accession number PRJEB15197. RNA-seq data have been deposited in the Gene Expression Omnibus, accession number GSE102870. The genome-wide association scan summary data will be made available at http://www.decode.com/summarydata.

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

## Acknowledgements

We thank the individuals who participated in this study and whose contributions made this work possible. We also thank our valued colleagues who contributed to the data collection and phenotypic characterization of clinical samples as well as to the genotyping and analysis of the whole-genome association data. This research has been conducted using the UK Biobank Resource under application number 24711. F.W.A. is supported by UCL Hospitals NIHR Biomedical Research Centre.

## Author contributions

G.R.O., A.O., M.K.M., D.F.G., U.T., P.S., and K.S. designed the study and interpreted the results. G.R.O., A.O., M.K.M., E.F., F.Z., A.H., J.S., G.T., F.W.A., V.T., T.R., H.H., I.O., P.T.O., G.L.N., and P.S. carried out the subject ascertainment and recruitment. G.R.O., G.H.H., G.A.A., A.M.K., A.L.L., As.J., Ad.J., S.J., R.L.G., B.G., and P.S. performed the sequencing, genotyping, and expression analyses. G.R.O., A.O., M.K.M., R.P.K., G.H.H., E.F., A.H., E.V.I., G.A.A., B.O.J., H.K., G.S., A.M.K., L.S., J.K.S., O.B.D., S.B., R.L.G., G.M., G.T., V.T., B.G., D.F.G., and P.S. performed the statistical and bioinformatics analyses. G.R.O., A.O., M.K.M., R.P.K., D.F.G., U.T., P.S., and K.S. drafted the manuscript. All authors contributed to the final version of the paper.

## Competing interests

Authors affiliated with deCODE genetics/Amgen Inc., G.R.O., A.O., M.K.M., R.P.K., G.H.H., E.F., F.Z., A.H., E.V.I., G.A.A., B.O.J., H.K., G.S., A.M.K., A.L.L., J.S., L.S., J.K.S., O.B.D., S.B., As.J., Ad.J., S.J., R.L.G., V.T., B.G., G.M., G.T., T.R., H.H., D.F.G., G.L.N., U.T., P.S., and K.S. declare competing interests as employees. The remaining authors declare no competing interests.
