## [Peer Review File · Communications Biology]

Reviewers' comments:

Reviewer #1 (Remarks to the Author):

Loss and gain of function mutations in ACO1 affect erythropoiesis

The authors combine UK Biobank with an Icelandic cohort (n~680K) to test for hemoglobin concentration genetic associations.

No discussion of what happened at the rest of the genome, just ACO1. Nor an explanation of why it was then focused on at a subthreshold level (e.g., rs147876514 has p=9e-04), nor any discussion on the implications of doing things this way.

Did you test previously reported hits for replication?

Heritability?

Methods state that the LD-Score correction factor was 0.68 in Iceland and 1.40 in UK Biobank. Is this the inflation factor (if not, also report the raw inflation factors), or the actual amount it was estimated that it needed to be corrected for? Much more description of what is going on here might be helpful. Why in particular is the Iceland one so off? The UK Biobank might be reasonable for a very polygenic trait. Are you trying to say that you deflated the UKB statistics, but you actually inflated the Iceland statistics?

Results first paragraph, 155,520 + 285,664 + 397,500 != 680,122 ?

Abstract, be clear, are all 8 novel?

Methods very cursory. Explain how did you trace the variants back to common ancestor born in the 18th century? UKB field codes used? How the UKB individuals were chosen for analysis? Etc.

Data availability section left completely blank. Are the authors planning on sharing summary statistics and making data available?

What, if anything, did the alternative prioritization gain you?

Reviewer #2 (Remarks to the Author):

In this paper by Oskarsson et al., the authors use genomic data from Iceland and the UK to identify mutations in alleles that affect Hgb levels. They find two mutant alleles in ACO1 that cause anemia in heterozygote carriers, and four that cause polycythemia. These findings are very exciting because they accord well with results predicted on the basis on the known bifunctional and mutually exclusive activities of ACO1, also known as Iron Regulatory Protein 1, which should be mentioned as a commonly used name for this gene product because it encompasses two activities, acting as a cytosolic aconitase in iron replete cells, and as a IRE binding protein known to repress HIF2 alpha, a factor for that enhances erythropoietin transcription. Some of the mutational effects can be readily explained by previous literature, but several are intriguing and it is not obvious how they would affect EPO, including Thr208A , R168W, and R802C.

The paper would be improved by a better description of the switch from cytosolic aconitase to IRE

binding. This is well described in Rouault, 2006, Nature Chem Biol.

Also, the wording on page 5 could be clearer- I would suggest saying..when iron is low(or NO high H2O2 high, , do not include hypoxia here, because it stabilizes rather than oxidizes the FeS cluster, and the effect of low iron is to increase the IRE binding activity of ACO1.. when iron increases, a FeS cluster binds ACO1 to yield a functional aconitase, which interconverts citrate and isocitrate in the cytosol and become inactive as an IRE binding protein due to a large conformational change. On page 7, after "lost its IRE binding activity, Walden and Volz, Science 2006 should be referenced. Also, in discussing the cytosolic aconitase structure, Dupuy, J., Volbeda, A., Carpentier, P., Darnault, C., Moulis, J. M., and Fontecilla-Camps, J. C. (2006). Crystal structure of human iron regulatory protein 1 as cytosolic aconitase. Structure 14, 129-139 should be referenced. Those mutations that cause polycythemia must cause reduced IRE binding activity of IRP1, though it is not obvious why that would be so in co-crystal structures. Lys 551 binds A15 of the IRE, as referenced in Walden, 2006. Notably, none of the 4 amino acid contacts to RNA bases that most adversely affect IRE binding, R269, K379, S371 and S681 were not affected in any of these pedigrees as described in Selezneva, A. I., Walden, W. E., and Volz, K. W. (2013). Nucleotide-specific recognition of iron-responsive elements by iron regulatory protein 1. J Mol Biol 425, 3301-3310. This is a very interesting paper that would benefit from a better narrative about why one protein could have two opposing roles, causing either anemia or polycythemia, depending on the mutation. it is also interesting that these mutations adversely affected heterozygotes. It was not easy to recognize which of the patients and mutations mentioned were found in homozygotes. The information was in the paper, but not readily accessible. I think the pedigree tables would be easier to follow if the carriers were half- red, and the obligate carriers were perhaps half red with hash marks. Many readers would assume that the full blocks represent homozygotes. Though the legend makes the point, it seems to defy convention. Please clarify which mutations had homozygote presentations for non-geneticists. The authors did a good job tying their findings to the discoveries in the literature, but some of the tables are quite oriented towards geneticists and would benefit from longer and clearer legends. Supplementary figures 4 and 5 could be more clearly presented.

Reviewer #3 (Remarks to the Author):

The paper reports results from association of two common and several rare variants in the ACO1 gene with hemoglobin concentration from a GWAS of over 600,000 individuals of the Iceland and UK Biobank datasets. This locus has been previously reported for hemoglobin and a common variant identified in this study explains the prior signal. Six rare coding variants were identified, including some variants imputed from a subset sample with whole exome sequencing from the Iceland data. The authors examined the association with erythrocyte and iron traits, specific for two rare variants identified in Iceland pedigrees (Cys506Ser, Lys334Ter). Lys334Ter was associated with transcript reduction in heterozygous carriers. The paper is well-written but mostly lack novelty. The main focus is on rare variants within one locus but it is not clear why they are reporting only variants at or nearby this gene. This also raises questions about the overlap of the reported results with previously published studies or other ongoing studies using these datasets. There is also some lack of clarity on the choice of the significance thresholds for p-values (including multiple thresholds) and how the study went from a discovery using combined Iceland/UK Biobank datasets to single study results for the ACO1 gene (Table 1).

Line 78, which common variant in Table 1 is the one that captures previously reported intergenic association and which methods were used to get to this conclusion.

Line 91, variants were also associated with RBC and HCT, which is not surprising as these are highly correlated phenotypes.

Paragraph starting on line 100: given the known biology for the gene protein in relation to

erythropoiesis, it would be of interest to examine the associations in the context of varying iron stores.

Paragraph starting in line 114, the title is misleading given the lack of functional studies for variants and assumptions are based on the direction of the beta estimates. For the sex-specific results, report the number of individuals with the alternative allele for rare variant within sex.

How relatedness was accounted for in analyses of Iceland participants?

There were 62 heterozygous for the Cys506Ser variant based on imputed data who clustered within a pedigree. Is there any other clinical consequence to the identified anemia in this pedigree? Tight this to the conclusion in relation to anemia of chronic disease.

Any evidence for selection at the region?

Line 265, give a more concrete example on how drug therapy targeting ACO1 can help in anemia due to chronic inflammation (see above question).

Line 285, for the multiple whole genome sequencing deCODE projects, were the datasets called and qc together?

Line 289, more details on the imputation are needed (minimal allele frequency, imputation quality).

Line 300, sample size for the UK Biobank does not match the sample used in this analysis. Exclusions, inclusion criteria need to be listed. Same for Iceland (line 326)

Paragraph starting on line 304, add number of measures used for mean hemoglobin for the individuals with the rare variants compared to those without them. Is there a difference in the period and number of measures between these groups? Can you also show the variance of the hemoglobin over time across these groups? Same for the iron measures including ferritin. Where the hemoglobin and iron measures collected at the same interval/time? What is the number of individuals with the alternative allele for rare variants that contributed data for ferritin, iron and other measures? Add some of this information to supplementary Table 4.

Line 331, not sure I understand inclusion of age of death as covariate in analyses of the Iceland dataset. This raises questions if the hemoglobin data was collected at the time of

illnesses/hospitalizations when acute anemia can occur, instead of steady state ambulatory setting.

Why adjust for country of origin? Principal components were not included in analyses and adjustments for relatedness are not described.

Supplementary Table suggest differences in the distribution of hemoglobin between Iceland and UK biobank. Given Iceland used multiple measures and UK biobank just one measure, are the scale of estimates equivalent for a meta-analysis using inverse variance methods?

Significance threshold paragraph (line 354) is not clear.

Point-by-point answer to reviewers

We would like to thank the three reviewers for their critique. We have made substantial changes to the manuscript in accordance to their comments and believe that we have addressed all their concerns.

All changes to the originally submitted manuscript can be reviewed in track-changes. Line numbers below refer to edits in the manuscript with track changes on.

Reviewers' comments

Reviewer #1 (Remarks to the Author):

Comment #1 a)

No discussion of what happened at the rest of the genome, just ACO1.

Response to Comment #1 a

In the current study we chose to focus on the biological effects of several variants in a single gene. It is unusual to observe several naturally occurring protein altering variants in the same gene with such a large opposing effects. These unique results at the *ACO1* locus deserve to be presented fully, and would not be done justice as a subsection in a large GWAS study cataloging all hemoglobin associating variants.

To address the reviewers concerns we will share summary statistics upon publication for GWAS of hemoglobin concentration in the Icelandic population.

In addition, we have added replication data on all reported GWAS associations with hemoglobin levels in Iceland and the UK Biobank (Supplementary Data 7) (see response to Comment #2).

To address the reviewers concerns we have added a paragraph to the Data availability section that reads (changes underlined)

Page 15 lines 419-425 under Data availability:

“Summary statistics for GWAS of hemoglobin concentration in the Icelandic population will be available for download upon publication from the GWAS Catalog FTP site ftp://ftp.ebi.ac.uk/pub/databases/gwas/summary_statistics/.”

Comment #1 b)

Nor an explanation of why it was then focused on at a subthreshold level (e.g., rs147876514 has $p=9e-04$), nor any discussion on the implications of doing things this way.

Response to Comment #1 b

There were five variants at *ACOI* associating with hemoglobin concentration at genome-wide significance, of which three are coding variants (Cys506Ser, Arg168Trp, and Thr208Ala) and two were common non-coding (rs12985, rs7045087). It is quite unusual to identify multiple independent variants in the same gene, especially when these variants are coding and have large opposing effects on a trait. Collectively these variants strongly implicate *ACOI* as an important regulator of hemoglobin concentration. Thus, we performed a secondary analysis to test all coding variants in the *ACOI* gene, and we chose a P-value threshold for the secondary analysis based on a Bonferroni correction for the number of coding variants in *ACOI* (N=34).

To address the concerns of the reviewer, we have added the following text to the introduction and discussion (changes are underlined).

Page 4 lines 77-82 in the results:

*“Five variants in *ACOI*, encoding cytosolic aconitase 1, also known as Iron-responsive element binding protein 1 (IRP1), associated genome-wide significantly with hemoglobin concentrations, of which three are coding and one common non-coding variant captures a previously reported intergenic association (Table 1). Subsequently, we tested the 34 remaining coding variants in *ACOI* for association with hemoglobin concentration and found three additional associations after accounting for multiple testing ($P < 0.05/34 = 1.5 \times 10^{-3}$) (Table 1 and Supplementary Data 1).”*

Page 9, lines 228-230 in the discussion:

*“The aim of this study is to understand how sequence variants in *ACOI* affect hematopoiesis. After we identified a genome wide significant association, we performed conditional analysis to identify secondary associations at the locus, focusing on variants with a predicted protein coding effect on *ACOI*.”*

Comment #2

Did you test previously reported hits for replication?

Response to Comment #2

We have added information on all reported GWAS associations with hemoglobin levels and how they replicate in Iceland and the UK Biobank. This information is now provided as supplementary material (supplementary data 7).

Page 8, lines 214-225, a new sub-section “Reported hemoglobin associated variants”:

“We show association results for the 138 reported associations of sequence variants with hemoglobin levels in populations of European descent, most of which (N = 119) come from the hereto largest hemoglobin GWAS reported by Astle et al. in 2016, where the UK biobank participated with 87 K individuals, which comprises 22% of the UK biobank dataset used in the current study (Supplementary data 7).

We replicated the hemoglobin association of 110 of the 138 variants (80%) at P-value below 0.05 and with a consistent direction of effect. When accounting for multiple testing ($P < 0.05/175 = 2.9 \times 10^{-4}$), 76 variants replicate in Iceland with a consistent direction of effect (Supplementary data 7).

For the combined Icelandic and UK datasets 135 out of 138 variants replicate at P-value of 0.05 and all show a consistent direction of effect (Supplementary data 7). When taking multiple testing into account, 133 out of the 138 hemoglobin levels associated variants replicate (Supplementary data 7).”

Comment #3

Heritability?

Response to Comment #3

To estimate the heritability of hemoglobin concentration, we calculated the SNP-heritability using LD score regression. We have now included these results as a supplementary table (Supplementary Table 2) and corresponding methods (Changes underlined).

Page 4, lines 75-76:

“Heritability of hemoglobin concentration in the Icelandic population was estimated to be 0.20 and 0.29 using parent-offspring and sibling correlations, respectively (Supplementary Table 2).”

And page 14, a new subsection at lines 372-375

“Heritability

Heritability of hemoglobin concentration was estimated in the following two ways: 1) $2 \times$ parent-offspring correlation, 2) $2 \times$ full sibling correlation, using the Icelandic data (where all family relationships are known).”

Comment #4

Methods state that the LD-Score correction factor was 0.68 in Iceland and 1.40 in UK Biobank. Is this the inflation factor (if not, also report the raw inflation factors), or the actual amount it was estimated that it needed to be corrected for? Much more description of what is going on here might be helpful. Why in particular is the Iceland one so off? The UK Biobank might be reasonable for a very polygenic trait. Are you trying to say that you deflated the UKB statistics, but you actually inflated the Iceland statistics?

Response to Comment #4

When testing association of sequence variants with quantitative traits we use a BOLT linear mixed model. These models are now widely used because they account for cryptic relatedness while also increasing power⁶.

One-step in the BOLT-LMM procedure (step 1b) is to calibrate the χ^2 test statistic by calculating a constant calibration factor c_{inf} in

$$\chi^2_{\text{BOLT-LMM-inf}} = \frac{(x_{\text{test}}' V_{\text{LOCO}}^{-1} y)^2}{c_{\text{inf}}}$$

To compute the calibration constant BOLT-LMM rapidly computes the prospective statistic at 30 random SNPs by applying conjugate gradient iteration.

We, however, did not apply this scaling to the test statistic in the Icelandic association model. Therefore, when we apply the LD score regression and estimate a correction factor from the regressions intercept it will be shifted by this constant factor. The intercept is therefore not comparable to factors obtained from standard genomic control methods, and should not be interpreted as such. It can indeed be below 1 due to the calibration factor.

If associations were computed under a standard generalized linear regression model the correction factor for the Icelandic data would be 1.57 which is like the reviewer points out reasonable for a very polygenic trait. We are therefore not inflating the Icelandic statistics when

applying genomic control. The text from the methods section where we state that the correction factor is 0.68 has now been removed in order to avoid confusion.

Sentence removed:

“The estimated correction factor for Hemoglobin concentration based on LD score regression was 0.68 for the additive model in the Icelandic sample and 1.40 in the UK Biobank.”

Comment #5

Results first paragraph, $155,520 + 285,664 + 397,500 \neq 680,122$?

Response to Comment #5

- 155,250 is the number of directly chip-typed Icelanders of which 143,682 have hemoglobin measurements.
- 285,664 is the number of individuals that are family imputed, i.e. genotype probabilities are calculated for untyped first and second degree relatives of chip-typed individuals based on Icelandic genealogy, of which 142,940 have hemoglobin measurements.
- For Iceland, we have genotype information on 440,914 ($155,250 + 285,664$) individuals. Of those, hemoglobin measurements are available for **286,622** ($143,682 + 142,940$) individuals.
- For the UK, hemoglobin measurements are available for all **397,500** genotyped individuals.
- In total 684,122 individuals with genotype and hemoglobin measurements make up the combined GWAS dataset, 286,622 from Iceland and 397,500.

For clarity, we have added the following text to the Results (changes are underlined).

Page 4, lines 64-72:

“In the meta-analysis we combined GWAS results on hemoglobin concentration from 286,622 Icelanders and 397,500 individuals from the UK (Supplementary Fig. 1, Supplementary Table 1). In Iceland, we tested 37.6 million sequence variants, identified through whole genome sequencing of 28,075 Icelanders and subsequently imputed into 155,250 chip-typed individuals, as well as 285,664 of their first- and second-degree relatives (imputation info > 0.8 and MAF > 0.01%). Out of a total of 440,914 individuals with genotype information, 286,622 have hemoglobin measurements available. In the UK, the GWAS was performed on 40 million markers (imputation info > 0.8), from the Haplotype Reference Consortium (HRC) reference panel, imputed into 397,500 chip-typed individuals of European ancestry from the UK Biobank and hemoglobin measurements were available for all.”

Note: The number of genotyped individuals in Iceland 282,622 was corrected to 286,622 in the paragraph above. This was due to a typo in the manuscript and the correct number is 286,622 as stated in the Methods.

Comment #6

Abstract, be clear, are all 8 novel?

Response to comment #6

Seven of the eight variants are novel, whereas rs7045087[C] was reported by Astle et al.¹, 2016. The abstract has been changed in accordance to the comment (changes underlined) page 2, lines 21-23.

“We performed genome-wide association studies of hemoglobin concentration using a combined set of 684,122 individuals from Iceland and the UK and found seven novel variants, six rare coding and one common, at the ACO1 locus. ”

Comment #7

Methods very cursory. Explain how did you trace the variants back to common ancestor born in the 18th century?

Response to Comment #7

Close to complete genealogical records of the Icelandic population are available dating back to the Icelandic national census of 1703, and incomplete records dating back to the settlement of Iceland in 874 CE^{9,10}. The Icelandic genealogy coupled with the large fraction of the population that has been chip typed allows the determination of the origin of sequence variants of chip-typed individuals through long-range phasing and haplotype imputation¹¹.

All carriers of the two rarest *ACOI* sequence variants, Cys506Ser and Lys334Ter, can be traced back to common ancestors in the genealogical records by identifying the most recent common ancestor of all carriers of each variant in the genealogical record.

Also, these rare variants are not found on the same haplotype background in any descendants of relatives of the common ancestor. Therefore it is highly likely that each of the mutations originated from a single common ancestor.

A description has now been added to Methods section

Page 13, lines 340-348.

“Determination of sequence variant origin

Close to complete genealogical records of the Icelandic population are available dating back to the Icelandic national census of 1703, and incomplete records dating back to the settlement of Iceland in 874 CE. The Icelandic genealogy coupled with the large fraction of the population that has been chip-typed allows us to determine the origin of sequence variants through long-range phasing and haplotype imputation. We used the Icelandic genealogy database to identify the most recent common ancestors of carriers of the two rarest ACOI sequence variants, Cys506Ser and Lys334Ter. In both cases, all carriers

shared a common ancestor. These sequence variants are absent from descendants of close relatives of the common ancestor carrying the same haplotype background.”

Comment #8

UKB field codes used? How the UKB individuals were chosen for analysis? Etc.

Response to Comment #8

We used UKB field code 30020 (Haemoglobin concentration). In the analysis we used 397,500 individuals of white British ancestry with hemoglobin concentration measurements. We have added this information to the Methods (changes underlined).

Page 13, lines 359-361;

“From the UK Biobank we used 418,628 hemoglobin concentration measurements from 397,500 individuals of white British ancestry, whose samples were collected at the UK Biobank assessment centers (Field ID 30020, Haemoglobin concentration) (Supplementary Table 10).”

Supplementary Table 10. UK Biobank field ID codes for quantitative tested in the current study.

Field ID code	Description
30020	Haemoglobin concentration
30010	Red blood cell (erythrocyte) count
30030	Haematocrit percentage
30000	White blood cell (leukocyte) count
30080	Platelet count
30250	Reticulocyte count
30060	Mean corpuscular haemoglobin concentration
30040	Mean corpuscular volume

Comment #9

Data availability section left completely blank. Are the authors planning on sharing summary statistics and making data available?

Response to Comment 9#

We have added text to the data availability section. The section now reads (changes underlined).

Page 15 lines 419-425 under Data availability:

“Summary statistics for GWAS of hemoglobin concentration in the Icelandic population will be available for download upon publication from the GWAS Catalog FTP site ftp://ftp.ebi.ac.uk/pub/databases/gwas/summary_statistics/.”

Comment #10

What, if anything, did the alternative prioritization gain you?

Response to Comment #10

The alternative prioritization gained us three protein coding variants in *ACO1*, in addition to the two genome-wide significant ones.

These five protein coding variants allowed us to put *ACO1* protein alterations into the context of existing literature on molecular biology of *ACO1* based on in cell- and animal models. Specifically, in the case of the Cys506Ser missense mutation we had the opportunity to study in humans the effects of a mutation that had been experimentally introduced into model systems to study the function of *ACO1*.

Reviewer #2 (Remarks to the Author):

Comment #1

The paper would be improved by a better description of the switch from cytosolic aconitase to IRE binding. This is well described in Rouault, 2006, Nature Chem Biol.

Response to Comment #1

We thank the reviewer for this comment and we have now added the following text to the discussion chapter (changes underlined)

Page 9, lines 240-249:

“The two variants in ACO1 with largest effects are both likely to have pronounced effects on protein function with the larger effect of Cys506Ser an order of magnitude larger than that of any previously reported sequence variants associating with decreased hemoglobin concentration: carriers have -1.61SD less hemoglobin, which corresponds to 24.6 g/L. This leads to a very high risk of persistent anemia among carriers (OR = 17.1). Structural studies have shown that when the [4Fe-4S] cluster is intact, protein domain 4 is folded over and covers the [4Fe-4S] cluster within the central core formed by domains 1 and 2. When the iron-sulfur cluster disassembles because of iron depletion (and/or because of oxidative degradation of the cluster) or when mutations in any of the [4Fe-4S] binding cysteines prevent cluster binding, domain 4 moves by a flexible hinge linker exposing the core domains. This allows the IRE structure to bind specifically to the protein.”

Comment #2

Also, the wording on page 5 could be clearer- I would suggest saying..when iron is low(or NO high H₂O₂ high, , do not include hypoxia here, because it stabilizes rather than oxidizes the FeS cluster, and the effect of low iron is to increase the IRE binding activity of ACO1.. when iron increases, a FeS cluster binds ACO1 to yield a functional aconitase, which interconverts citrate and isocitrate in the cytosol and become inactive as an IRE binding protein due to a large conformational change.

Response to Comment #2

We thank the reviewer for pointing this out. The wording on page 5 has been made clearer in accordance to the reviewers suggestions (changes underlined).

Page 5, lines 111-116:

“When iron is low (or NO high, H₂O₂ high), the IRE binding activity of ACO1 increases and it binds iron responsive element (IRE) in the 5’ and 3’ untranslated region of mRNAs of many genes involved in iron regulation. When the concentration of iron increases, an [4Fe-4S] cluster binds to ACO1 to yield a functional aconitase, which interconverts citrate and isocitrate in the cytosol and becomes inactive as an IRE binding protein due to a large conformational change.”

Comment #3

On page 7, after "lost its IRE binding activity, Walden and Volz, Science 2006 should be referenced.

Response to Comment #3

We thank the reviewer for pointing this out. Walden and Volz (2006) is now referenced as suggested and is now reference no. 25 and is cited at page 7, line 181.

Comment #4

Also, in discussing the cytosolic aconitase structure, Dupuy, J., Volbeda, A., Carpentier, P., Darnault, C., Moulis, J. M., and Fontecilla-Camps, J. C. (2006). Crystal structure of human iron regulatory protein 1 as cytosolic aconitase. Structure 14, 129-139 should be referenced.

Response to Comment 4#

We thank the reviewer for pointing this out. Dupuy et al. (2006) is now referenced as suggested and is now reference no. 29 and is cited at page 9, lines 246, 249, and 262.

Comment #5

Those mutations that cause polycythemia must cause reduced IRE binding activity of IRP1, though it is not obvious why that would be so in co-crystal structures. Lys 551 binds A15 of the IRE, as referenced in Walden, 2006. Notably, none of the 4 amino acid contacts to RNA bases that most adversely affect IRE binding, R269, K379, S371 and S681 were not affected in any of these pedigrees as described in Selezneva, A. I., Walden, W. E., and Volz, K. W. (2013). Nucleotide-specific recognition of iron-responsive elements by iron regulatory protein 1. *J Mol Biol* 425, 3301-3310.

Response to Comment 5#

The point that the reviewer is making is interesting and we now discuss it in the manuscript. We have added the following text in the discussion (changes underlined).

Page 10, lines 270-274:

“We speculate that the coding variants associated with increased hemoglobin concentration likely reduce the IRE binding activity of ACO1, though it is not clear how that would happen based on co-crystal structures. None of the four coding variants identified are in close proximity to the amino acids known to most adversely affect IRE binding: Arg269, Lys379, Ser371, and Ser681. However, Asn549Ile is close to Lys551, which binds A15 of the IRE.”

Comment #6

This is a very interesting paper that would benefit from a better narrative about why one protein could have two opposing roles, causing either anemia or polycythemia, depending on the mutation. it is also interesting that these mutations adversely affected heterozygotes.

Response to Comment #6

We thank the reviewer for pointing this out. To make these messages clearer for the reader we have changed the final paragraph in the Discussion section,

Pages 10-11, lines 284-303, which now reads (changes underlined):

*“Finding several variants in the same gene that affect the function of the protein it encodes can lead to a better understanding of the role of the protein in both normal and abnormal biology. Here we report sequence variants with both loss or gain-of-function effects on the same gene. Loss-of-function variants allow the identification of processes for which a gene is required, while gain-of-function variants indicate that the gene is able to control the process it affects in a rheostatic manner. The effects the *ACO1* variants have on hemoglobin and ferritin, either increasing or decreasing levels, suggest a regulatory function of *ACO1* with effects that go both ways. The effects of the loss-of-function variants reported here most likely result from *ACO1* haploinsufficiency, as we demonstrate for the stop-gained heterozygotes for *Lys334Ter*. The underlying mechanism of gain-of-function variants are usually harder to explain. In case the of the *Cys506Ser* variant, the mechanism is well studied in model systems and is the result of a gain-of IRE-binding. Other coding variants in *ACO1* that produce similar phenotypic effects are most likely to go through the same mechanism of action. Both loss- and gain-of-function variants in *PCSK9* have been identified that decrease and increase cholesterol levels, respectively, and led to the development of *PCSK9* inhibitors to reduce LDL cholesterol levels. Also, identification of loss- and gain-of-function variants in *SCN9A* encoding a voltage-gated sodium channel cause insensitivity to pain (recessive) and paroxysmal*

extreme pain disorder (dominant), have triggered efforts to develop SCN9A inhibitors as a therapeutic. The identification of loss- and gain-of-function variants in ACO1 sheds light on mechanisms that could be exploited in the development of therapies targeting erythropoiesis.”

Comment #7a

It was not easy to recognize which of the patients and mutations mentioned were found in homozygotes. The information was in the paper, but not readily accessible.

Response to Comment #7a

To address the concerns of the reviewer we have made a table showing hetero- and homozygous counts in the Iceland and UK datasets for the minor allele of the eight *ACO1* variants reported in the manuscript (Supplementary Table 7).

Supplementary table 7 is now referenced to at page 5, line 126:

“In Iceland, one in 2,600 individuals are heterozygous for Cys506Ser (Supplementary Table 7)”,

Supplementary Table 7. Counts of heterozygotes and homozygotes for the minor alleles of the ACO1 variants in Iceland and UK datasets. Amin: Minor allele; Amaj: Major allele; MAF: Minor Allele Frequency; Consequence: consequence of sequence variants on transcript or protein level; N het.: number of heterozygous carriers; N homo.: number of homozygous carriers of the corresponding variant.

Position (Hg38)	rs name	Amin/A maj	MAF Ice/UK (%)	Consequence	Iceland		UK	
					N het.	N homo.	N het.	N homo.
chr9:32429450	-	A/T	0.02/-	Cys506Ser	62	0	-	-
chr9:32450189	rs12985	C/T	35.9/37.1	*78T>C	70,198	19,611	187,883	54,978
chr9:32455264	rs7045087	C/T	27.6/30.0	Intergenic	60,530	11,801	169,281	35,318
chr9:32418355	rs41305321	T/C	0.48/0.12	Arg168Trp	1,458	7	1090	0
chr9:32430494	rs750337798	T/A	0.21/-	Asn549Ile	616	0	-	-
chr9:32418475	rs61753543	G/A	0.16/0.12	Thr208Ala	485	1	653	0
chr9:32423348	rs745558996	T/A	0.02/-	Lys334Ter	67	0	-	-
chr9:32448929	rs147876514	T/C	-/0.01	Arg802Cys	-	-	65	0

Comment #7b

I think the pedigree tables would be easier to follow if the carriers were half- red, and the obligate carriers were perhaps half red with hash marks. Many readers would assume that the full blocks represent homozygotes. Though the legend makes the point, it seems to defy convention. Please clarify which mutations had homozygote presentations for non-geneticists.

Response to Comment #7b

To address the concerns of the reviewer we have remade Figure 1 and Figure 3 according to the suggestions made by the reviewer. To adhere to convention regarding pedigree figures heterozygotes are indicated with half filled symbols and obligate carriers are indicated with a dot.

Figure 1. Pedigree of carriers of Cys506Ser in ACO1. All 62 genotyped carriers can be traced back to ancestors born in the late 18th century. Shown is a shortest path pedigree. The founding couple had eight offspring, two of which were carriers of the Cys506Ser missense variant, and a current total number of 5,430 descendants. Roman numerals indicate generation, mean hemoglobin concentration is noted below the symbols. square = male, circle = female, diamond = sex unspecified, solid half-filled object = carrier, dotted object = obligate carrier, red filled object = persistent anemia.

Figure 3. Pedigree of carriers of Lys334Ter in ACO1. All 67 genotyped carriers can be traced back to a common ancestor in the early 18th century. Shown is a shortest path pedigree. The founding couple had six offspring two of which were carriers of the Lys334Ter variant, and a current total number of 21,423 descendants. Roman numerals indicate generation, year of birth of the founding couple is noted above the symbols and mean hemoglobin concentration is noted below the symbols. square = male, circle = female, diamond = sex unspecified, solid filled object = carrier, half filled object = obligate carrier, red filled object = polycythemic. Obligate carrier status is not indicated before generation VII as no phenotype information is available for those individuals.

Comment #8a

The authors did a good job tying their findings to the discoveries in the literature, but some of the tables are quite oriented towards geneticists and would benefit from longer and clearer legends. Supplementary figures 4 and 5 could be more clearly presented.

Response to Comment #8a

We thank you for pointing this to us. We have now make changes to table legends for the following tables and they now read (changes underlined):

Table 1. *Variants in ACO1 associating with hemoglobin concentration in the meta-analysis of the Icelandic and the UK datasets. Effect is shown for the minor allele in standard deviations. Significance levels and effects are shown for the combined analysis. Amin: Minor allele; Amaj: Major allele; MAF: Minor Allele Frequency; Consequence: consequence of sequence variants on transcript or protein level (NM_001278352.1 and NP_001265281.1) according to HGVS nomenclature; LD:Linkage disequilibrium; LD-class size: total number of variants correlating with $R^2 > 0.8$ to the variant; P-het: P-value for test of heterogeneity of effect between Iceland and UK.*

Table 2. *Associations of variants in ACO1 and other relevant hematological quantitative phenotypes in the Icelandic-UK meta-study. N is the number of individuals measured for each parameter. Effect is shown in standard deviations for the minor allele. Significance levels and effects are shown for the combined analysis. HGVS is definition the mutation according to the Human Genome Variation Society nomenclature. MCV: Mean Corpuscular Volume; WBC: White Blood Cell Count; PLT: platelets; IBC: Iron Binding Capacity; Tf sat = Transferrin Saturation.*

Table 3. *Associations of variants in ACO1 and relevant hematological case-control phenotypes in the Icelandic-UK meta-study. Persistent anemia is where an individual has all hemoglobin concentration measurements below defined*

threshold of anemia based on gender. The polycythemia phenotype was defined as individuals that were at least once measured to be above the defined threshold of polycythemia based on gender. Controls for both phenotypes were individuals never reaching the hemoglobin threshold level for definition of the phenotype based on gender. N cases is the number of individuals defined to have the phenotype based on hemoglobin measurements (methods). N controls is the number of individuals that do not fulfill the criteria to be defined with the phenotype. Effect is shown in odds ratio for the minor allele. Significance levels and effects are shown for the combined analysis. HGVS is definition the mutation according to the Human Genome Variation Society nomenclature. OR = Odds Ratio

Supplementary Table 1. Mean and standard deviations of hemoglobin concentration in Iceland and in the UK stratified by gender. N is the number of individuals. Mean is the mean of the mean hemoglobin concentration measurements for each individual. SD: Standard deviation.

Supplementary Table 3. Conditional analysis of the ACO1 variants in the Icelandic dataset, where each variant is tested with all the other variants as covariates. All coding variants correlated with $r^2 < 0.009$ to one another based on linkage disequilibrium (LD), however, the two common non-coding variants, rs12985 and rs7045087, correlated with $r^2 = 0.13$. Effect is shown for the minor allele in standard deviations. HGVS: Human genome variation society nomenclature; LD-class: total number of variants correlating with $R^2 > 0.8$ to the variant (stratified by functional impact class, where HIGH impact variants include stop-gained, frameshift, splice acceptor or donor; MODerate impact variants include missense, splice-region variants and in-frame indels; LOW impact variants include upstream and downstream variants; and LOWEST impact variants include intron and intergenic variants); Amin: Minor allele; Amaj: Major allele

Supplementary Table 4. Comparison of measurements for hemoglobin concentration, iron, and ferritin between carriers of the two rarest ACO1 variants (Lys334Ter and Cys506Ser) and non-carriers. Percentiles of age of measure (AOM) are shown. Mean: The mean value of measurement for corresponding hematological measurement; SD: Standard Deviation; AOM: Age of measure)

Supplementary Table 5. Comparison of hemoglobin concentration, iron, and ferritin levels between different age groups for carriers of the two rarest ACO1 variants (Lys334Ter and Cys506Ser) and non-carriers. AOM: Age of measure; n: The number of individuals within corresponding age group; mean: The mean value of measurement for corresponding hematological measurement; sd: Standard Deviation.

Supplementary Table 6. Association of variants in ACO1 with hemoglobin concentration and ferritin. Effect is shown for the minor allele in standard deviations. Significance levels and effects for hemoglobin are shown for the combined analysis. Ferritin measurements were only available from the Icelandic dataset. Amin: Minor allele; Amaj: Major allele; HGVS: Human genome variation society nomenclature; SD: standard deviation.

Supplementary Table 7. Counts of heterozygotes and homozygotes for the minor alleles of the ACO1 variants in Iceland and UK datasets. Amin: Minor allele; Amaj: Major allele; MAF: Minor Allele Frequency; Consequence: consequence of sequence variants on transcript or protein level; N het.: number of heterozygous carriers; N homo.: number of homozygous carriers of the corresponding variant.

Davydov et al. 2010 (PLoS Comput. Biol) has been now referenced in **supplementary table 8.**

Supplementary Table 9. Association of variants in ACO1 with ACO1 expression based on RNA-sequencing data from 13,163 Icelanders. N is the number of

individuals. Effect is shown for the minor allele in standard deviations. Amin: Minor allele; Amaj: Major allele; Consequence: consequence of sequence variants on transcript or protein level; HGVS_p: Human genome variation society nomenclature; SD: standard deviation; N Controls: number of controls (non-carriers); N hetero.: number of heterozygous carriers; N homo.: number of homozygous carriers of the corresponding variant.

Supplementary Table 10. UK Biobank field ID codes for quantitative tested in the current study.

Comment #8b

Supplementary figures 4 and 5 could be more clearly presented.

Response to Comment #8b

For Supplementary figure 4, the figure the font size has been increased and the unit of measurement has been added to the axis labels.

Supplementary figure 4. A scatter plot showing the two common and the five ACOI rare coding variants Arg168Trp, Thr208Ala, Lys334Ter, Cys506Ser and Asn549Ile. The x-axis shows the effect on ferritin levels in Iceland while the y-axis shows the effect on hemoglobin concentration in the Iceland + UK meta-analysis. For each variant the allele associating with increased ferritin was selected and effect shown in standard deviations (SD). Correlation=0.95 (95%CI: 0.38-1.00), $p=0.015$

For Supplementary figure 5 the font size has been increased and legend position was changed to increase readability. Changes were also made to the figure caption for clarity.

Supplementary Figure 5. A) Mean hemoglobin concentration of 46 carriers of *p.Cys506Ser* in ACOI ($n_{\text{males}} = 20$, $n_{\text{female}} = 26$) in comparison to hemoglobin concentration of non-carriers ($n_{\text{total}} = 138,453$) aged 11 to 85 years old. *Cys506Ser* associates with decreased hemoglobin concentration (effect = -1.61SD , corresponding to 24.6 g/L , $P = 2.6 \times 10^{-24}$). A significant difference in effect is not observed between genders when tested separately (heterogeneity $P = 0.59$). B) Mean hemoglobin concentration of 61 carriers of *p.Lys334Ter* in ACOI ($n_{\text{males}} = 34$, $n_{\text{female}} = 28$) in comparison to hemoglobin concentration of directly genotyped non-carriers ($n_{\text{total}} = 138,453$) aged 11 to 85 years old. *Lys334Ter* associates with increased mean hemoglobin concentration compared to non-carriers (Effect = 0.63SD , corresponding to 9.7 g/L $P = 6.1 \times 10^{-6}$). A significant difference in effect is not observed between genders when tested separately (heterogeneity $P = 0.57$). Number of individuals representing each graphical point are indicated above the point and confidence intervals are depicted as gray vertical lines.

Reviewer #3 (Remarks to the Author):

Comment #1a

The main focus is on rare variants within one locus but it is not clear why they are reporting only variants at or nearby this gene.

Response to Comment 1#a

We chose to focus on the biological effects of several variants in *ACOL1*. It is unusual to observe several naturally occurring protein altering variants in the same gene with large opposing effects. These unique results deserve to be presented fully, and would not be done justice as a subsection in a large GWAS study cataloging all hemoglobin associating variants.

See response to Reviewer #1 comment #1a for a more comprehensive response

Comment #1b

This also raises questions about the overlap of the reported results with previously published studies or other ongoing studies using these datasets.

Response to Comment #1b

The Icelandic dataset has not been used in previous GWAS of hemoglobin concentration.

A fraction of the UK biobank dataset was used in the study by Astle et al. where the UK biobank participated with 87 K individuals¹ which comprises 22% of the UK biobank dataset used in the current study. Thus 78% have not been published previously.

The only previously reported marker reported in the current study is the common variant rs12985¹.

It is however difficult to speculate on other studies. Clearly the UK Biobank is public and researchers can freely access that data.

To demonstrate that the datasets used in the current study are consistent with previous reports we now provide replication of previous reports (see response to Reviewer #1 comment #2).

In addition, to allow other researchers to benefit from our efforts we will share summary statistics down for hemoglobin concentration upon publication.

Comment #2

There is also some lack of clarity on the choice of the significance thresholds for p-values (including multiple thresholds) and how the study went from a discovery using combined Iceland/UK Biobank datasets to single study results for the ACO1 gene (Table 1).

Response to Comment #2

See response to Reviewer #1 comment #1b

Comment #3

Line 78, which common variant in Table 1 is the one that captures previously reported intergenic association and which methods were used to get to this conclusion.

Response to Comment #3

The variants rs7045087 is the same variant that was reported by Astle et al. at this locus. We came to this conclusion by referencing the GWAS catalog database and checking the original publication.

- The text in lines 80-81 (previously line 78) now reads
“...of which three are coding and one common non-coding variant rs7045087 represents a previously reported intergenic association (Table 1).”
- In table 1 the reported variant rs7045087 is marked with a footnote (+) that reads “Previously reported in Astle, 2016”.

Comment #4

Line 91, variants were also associated with RBC and HCT, which is not surprising as these are highly correlated phenotypes.

Response to Comment #4

We agree with the reviewer that hemoglobin concentration, RBC and HCT are all highly correlated phenotypes. Any one of which could have been chosen as the main phenotype of the current study. In the manuscript we address this in the third paragraph of the Results section (changes underlined).

Page 4 and lines 94-98:

“All eight variants associating with hemoglobin also associate with red blood cell counts (RBC) and hematocrit (HCT) with similar significance, direction and magnitude of effect, consistent with the high correlation of the three phenotypes. Hemoglobin concentration was used as the primary GWAS phenotype and the correlated phenotypes for lookup.”

Comment #5

Paragraph starting on line 100: given the known biology for the gene protein in relation to erythropoiesis, it would be of interest to examine the associations in the context of varying iron stores.

Response to Comment #5

In order to investigate iron storage, we tested the variants for association with iron, ferritin, iron binding capacity, and transferrin saturation in the Icelandic population (Table 2 and Supplementary Data 5). Only the Cys506Ser missense variant associates with reduced ferritin levels in the Icelandic dataset. The red blood cell indices associated with the *ACO1* variants do not indicate that the effects on hemoglobin are driven by changes in iron stores. However, we cannot exclude the possibility that the *ACO1* variants are associated with subtle abnormalities in iron homeostasis.

Comment #6

Paragraph starting in line 114, the title is misleading given the lack of functional studies for variants and assumptions are based on the direction of the beta estimates.

Response to Comment #6

In case the of the Cys506Ser variant, the mechanism is well studied in model systems and is the result of a gain-of IRE-binding. For the other coding variant, Thr208Ala, which shows the same direction of effect, we are making the assumption that it is going through the same mechanism of action based on the known mechanism of the Cys506Ser variant. In the text we are careful to state that this mechanism of action is suggestive and is based on effect estimates.

We have changed the title to (changes underlined), page 5, line 119:

“Predicted RNA binding gain-of-function variants”

Comment #7

For the sex-specific results, report the number of individuals with the alternative allele for rare variant within sex.

Response to Comment #7

We thank the reviewer for pointing it out. We have adjusted the text by adding the suggested information.

Page 5, lines 124-125 now reads:

“We observed no difference in effect for hemoglobin concentration between male and female carriers of Cys506Ser ($P = 0.59$, $n_{\text{males}} = 20$, $n_{\text{females}} = 26$) (Supplementary Fig. 5A)”

Comment #8

How relatedness was accounted for in analyses of Iceland participants?

Response to Comment #8

The study is based on a large set of individuals who represent a large fraction of a founder population. Our method of testing for association takes the closest relatedness into account using a mixed effect model implemented in BOLT-LMM⁶.

See also Response to Reviewer #2 Comment #4 and the “Association analysis” subsection in the Methods page 14, lines 389-394

“For the meta-analysis we used a fixed-effects inverse variance method based on effect estimates and standard errors from the Icelandic and the UK Biobank study. For each study we used linkage disequilibrium (LD) score regression to account for distribution inflation in the dataset due to cryptic relatedness and population stratification. Using a set of about 1.1 million sequence variants with available LD score, we regressed the χ^2 statistics from our GWAS scan against LD score and used the intercept as correction factor.”

Comment #9

There were 62 heterozygous for the Cys506Ser variant based on imputed data who clustered within a pedigree. Is there any other clinical consequence to the identified anemia in this pedigree? Tight this to the conclusion in relation to anemia of chronic disease.

Response to Comment #9

We performed a phenome-wide scan testing the Cys506Ser missense variant in *ACOI* for association with 396 binary phenotypes in Iceland. No significant association with binary phenotypes was observed in addition to persistent anemia as reported in the manuscript when taking the number of tested phenotypes into account ($P < 0.05/396 = 1.3 \times 10^{-4}$).

We have now adjusted the text by adding the suggested information.

Pages 5-6, lines 130-136 now read:

“Consistent with the large effect on hemoglobin concentration, we detect an association of Cys506Ser with a high risk of persistent anemia (all hemoglobin measurements < 118 g/L for women and < 134 g/L for men) (Table 3). Persistent anemia was observed in 15 (28.3 %) of the 53 Cys506Ser carriers with hemoglobin measurements compared to 1.7% of the population (OR = 17.1, $P = 2.0 \times 10^{-14}$). We do not observe significant association with other disease phenotypes in the Icelandic population, given the number of phenotypes tested (significance threshold: $P < 0.05/396 = 1.3 \times 10^{-4}$) (Supplementary Data 6).”

Comment #10

Any evidence for selection at the region?

Response to Comment 10#

To our knowledge the *ACOI* locus has not been reported to be under selection¹⁶.

In the gnomAD database *ACOI* does not show signs of selective constraint as measured by per-gene constraint scores. *ACOI* has a Z-score of 0.92 for missense variants that indicates that the observed count of missense variation in the gene does not deviate significantly from what would be expected under neutral selection. The pLI (probability of loss-of-function intolerance) score for *ACOI* is 0 indicating that loss-of-function variation in the gene is tolerated. This data is consistent with our observations that coding variation in *ACOI* has moderate impact on health and is unlikely to affect reproductive success. At least at the heterozygous level the rarest variants with the largest effects, Cys506Ser and Lys334Ter, are observed within extended pedigrees, and are thus compatible with reproduction. However, we observed no homozygotes for these variants which is consistent with their rarity and we can therefore not conclude anything when it comes to the effect of these variants on homozygotes.

Comment #11

Line 265, give a more concrete example on how drug therapy targeting ACO1 can help in anemia due to chronic inflammation (see above question).

Response to Comment #11

As we have substantially deepened the discussion on the biological and genetic effects of *ACO1* variants (see responses to Reviewer #2, comments #1 & #6) we have decided to delete the sentence on anemia of inflammation (*"Many chronically anemic patients, such as those with anemia of inflammation, do not respond well to EPO"*). We furthermore concede that this suggestion is highly speculative and should thus be deleted.

Comment #12

Line 285, for the multiple whole genome sequencing deCODE projects, were the datasets called and qc together?

Response to Comment #12

The whole-genome sequenced samples were variant called jointly and the sequence variants found through whole-genome sequencing were phased jointly. The long-range phasing and imputation steps are performed for all chip-typed individuals, participating in various disease projects at deCODE Genetic, simultaneously¹¹. This approach incorporates many different quality controls to overcome batch effects and provides accurate genotype-calling – in particular, it leverages long-range haplotype sharing to validate genotype calls. In our data, we observe a high imputation accuracy, where 96.7% of variants with a minor allele frequency over 0.1 % achieve an imputation information over 0.8 which is beneficial when imputing with whole-genome sequencing data. Furthermore Gudbjartsson et al, 2015, found that the concordance for 28,204 chip-typed SNPs, which were not part of the long-range phasing set, was high (98.4% of SNPs with DAF >1% were imputed accurately, $r^2 > 0.9$)¹¹.

To address the reviewers concerns we have added a sentence to the Genotyping subsection of the Online Methods. .

Page 12, lines 328-333 now read (changes underlined):

“In addition, genotype probabilities for 285,644 untyped close relatives of chip-typed individuals were calculated based on Icelandic genealogy. The whole-genome sequenced samples were variant called jointly and the sequence variants found through whole-genome sequencing were phased jointly. The process used for whole-genome sequence sequencing of Icelanders, and the subsequent imputation from which the data for this analysis were generated has been extensively described in recent publications.”

Comment #13

Line 289, more details on the imputation are needed (minimal allele frequency, imputation quality).

Response to Comment #13

Details on the imputation have been added to the sentence.

Page 12, lines 326-328 now read (changes underlined):

“The chip-typed individuals were long range phased, and the variants identified in the whole-genome sequencing of Icelanders were imputed into the chip-typed individuals (imputation info > 0.8 and MAF > 0.01%).”

Comment #14

Line 300, sample size for the UK Biobank does not match the sample used in this analysis. Exclusions, inclusion criteria need to be listed. Same for Iceland (line 326)

Response to Comment #14

In the UK Biobank there are 408,658 individuals with imputed genotype. Of those, 397,500 had available hemoglobin concentration measurements. For UK Biobank, we used individuals of white British ancestry with hemoglobin concentration measurements in the analysis. For Iceland, we used all Icelandic individuals with available genotype and hemoglobin concentration measurement that had signed informed consent.

See also comment #8 Reviewer #1

Comment #15a

Paragraph starting on line 304, add number of measures used for mean hemoglobin for the individuals with the rare variants compared to those without them. Is there a difference in the period and number of measures between these groups?

Response to Comment #15a

To address the reviewers comment Supplementary Table 4 now includes the number of measurements and Supplementary Table 5 shows age distribution for carriers of the two rarest *ACOI* variants (Lys334Ter and Cys506Ser) and non-carriers.

Supplementary tables 4 and 5 are now referred to at Page 4, line 93 and page 13 lines 352-353 :

“...seven blood cell indices and five iron biomarkers (Supplementary tables 4 and 5),...”

We also note that effect on hemoglobin concentration reported in the current study are adjusted for age.

Comment #15b

Can you also show the variance of the hemoglobin over time across these groups? Same for the iron measures including ferritin.

Response to Comment #15b

For clarity we have also added Supplementary Table 5 that shows mean hemoglobin concentration, variance and number of measurements for different age groups for carriers of the two rarest *ACOI* variants (Lys334Ter and Cys506Ser) and non-carriers.

Comment #15c

Where the hemoglobin and iron measures collected at the same interval/time? What is the number of individuals with the alternative allele for rare variants that contributed data for ferritin, iron and other measures? Add some of this information to supplementary Table 4.

Response to Comment #15c

Iron concentration measurements were only available for the Icelandic dataset. In Iceland, roughly 5 times as many hemoglobin concentration measurements are available per individual compared to iron concentration (25.6 for hemoglobin vs. 4.9 for iron). For each individual we use the mean of all available measurements in the association analysis, where the measurements
To address the reviewers concerns we now have updated Supplementary Table 4 to include information about individuals contributing to the data of ferritin and iron. We also made Supplementary Table, which 5 shows the number of measurements of different hematological parameters for age groups for carriers of the two rarest ACO1 variants (Lys334Ter and Cys506Ser) and non-carriers.

Supplementary Table 4. Comparison of measurements for hemoglobin concentration, iron, and ferritin between carriers of the two rarest ACO1 variants (Lys334Ter and Cys506Ser) and non-carriers. Percentiles of age of measure (AOM) are shown. Mean: The mean value of measurement for corresponding hematological measurement; SD: Standard Deviation; AOM: Age of measure).

Carrier-status	Hemoglobin (g/L)			Iron ($\mu\text{mol/L}$)			Ferritin ($\mu\text{g/L}$)		
	Cys506 Ser	Lys334 Ter	Non-carriers	Cys506 Ser	Lys334 Ter	Non-carriers	Cys506 Ser	Lys334 Ter	Non-carriers
N individuals	53	63	286,506	27	22	123,640	34	32	172,918
N meas.	1,164	1,532	7,329,941	240	93	605,098	339	126	952,427
Mean N meas.	25.9	24.3	25.6	8.9	4.2	4.9	10	3.9	5.5
Mean	120	145	135	14.4	17	15.8	68.2	146	104
SD	12.2	14.3	15.7	4.2	8.3	6.5	55.5	122	93.2
AOM 25%	65	57	45	64	53	39	60	40	37
AOM 50%	72	68	63	70.5	72	55	70	69	53
AOM 75%	78	77	76	79	78	72	79	78	70

Supplementary Table 5. Comparison of hemoglobin concentration, iron, and ferritin levels between different age groups for carriers of the two rarest ACO1 variants (Lys334Ter and Cys506Ser) and non-carriers. AOM: Age of measure; n: The number of individuals within corresponding age group; mean: The mean value of measurement for corresponding hematological measurement; sd: Standard Deviation.

	AOM	Non-carrier			Cys506Ser			Lys334Ter		
		n	mean	sd	n	mean	sd	n	mean	sd
Hemoglobin	<20	87,414	131	14.6	7	129	12.9	2	149	20.3
	21-40	114,201	137	15.5	19	120	15.1	26	146	17.1
	41-60	107,031	139	13.3	21	121	11.1	33	149	11.6
	61-80	79,314	135	14.4	19	119	9.0	27	143	14.5
	81-100	28,755	125	14.6	6	111	7.5	7	133	11.6
	>100	187	118	14.0	0	-	-	0	-	-
Iron	<20	23,076	14.5	6.8	1	11		0	-	-
	21-40	35,126	16.7	6.9	11	12.8	5.5	6	18.9	10.5
	41-60	40,273	16.9	6.3	14	14.8	3.1	11	19.1	8.1
	61-80	33,831	15.7	6.3	10	15.2	2.4	8	13.1	5.4
	81-100	14,075	12.7	5.8	6	17.2	2.8	2	14.6	1.9
	>100	45	9.6	4.5	0	-	-	0	-	-
Ferritin	<20	33,522	46	39.2	2	6.5	0.71	1	57	
	21-40	570,60	73	73.2	13	61	60.6	9	118	134.0
	41-60	60,016	114	96.1	17	52	31.8	12	173	149.1
	61-80	44,911	151	102.6	16	74	37.0	10	134	61.8
	81-100	16,284	155	108.4	6	179	161.5	5	179.7	133.2
	>100	47	147	101.8	0	-	-	0	-	-

Comment #16a

Line 331, not sure I understand inclusion of age of death as covariate in analyses of the Iceland dataset. This raises questions if the hemoglobin data was collected at the time of illnesses/hospitalizations when acute anemia can occur, instead of steady state ambulatory setting.

Response to Comment #16a

In Iceland the hemoglobin concentration measurements were obtained from hospital records independent of patient status. For patients with multiple measurements we used the mean value in the analysis. The nature of the data and its origin is such that most measurements are from ambulatory setting, since hemoglobin is always measured in blood tests. We adjusted for time to death to correct for anemia because of illness. We used the mean value for those with multiple measurements in the analysis, which should decrease the effects of illnesses and hospitalizations on hemoglobin concentration.

Comment #16b

Why adjust for country of origin?

Response to Comment #16b

Iceland is historically divided into 23 counties corresponding to different geographical regions of the island (https://en.wikipedia.org/wiki/Counties_of_Iceland). We adjust for county (i.e. not country) of origin within Iceland which would take geographical variation in sequence diversity in Iceland into account.

See response to comment 16c# below for a more detailed discussion on the topic.

Comment #16c

Principal components were not included in analyses and adjustments for relatedness are not described.

Response to Comment #16c

In the Icelandic population the first principal components correlate strongly with the county of birth^{17,18}. However, they account for a very small proportion of genotypic variance¹⁸. This suggests that the effect of adjusting for these components will be very small. For the Icelandic dataset county of origin is included as a covariate in the logistic regression model and corrected for the QT analysis⁶ (See “Association analysis” subsection in Methods).

For information on adjustments for relatedness see response to Reviewer #3 comment #8

For information on the UK dataset see the “Association analysis” subsection in the Methods page 14, lines 386-387::

“In the UK Biobank study, 40 principal components were used to adjust for population stratification and age and sex were included as covariates in the logistic regression model and the BOLT-LMM.”

Comment #17

Supplementary Table suggest differences in the distribution of hemoglobin between Iceland and UK biobank. Given Iceland used multiple measures and UK biobank just one measure, are the scale of estimates equivalent for a meta-analysis using inverse variance methods?

Response to Comment #17

The Icelandic measurements were standardised prior to taking the average over multiple measurements, the same way as for the UK biobank data. Hence the measurements in Iceland and UK biobank are on the same scale (in normalized standard deviations). However, as shown in Table 1, the variance of the measurements is larger in the Icelandic data compared the UK biobank data. The reason is probably that most of the Icelandic measurements come from hospitals and therefore include many patients that have abnormal hemoglobin levels. As the distributions are standardized, this would affect the effect estimates, leading to lower effect estimates in the Icelandic GWAS.

Comment #18

Significance threshold paragraph (line 354) is not clear.

Response to Comment #18

Significance threshold paragraph now reads

Pages 14-15, lines 396-402 (changes underlined):

“We applied genome-wide significance thresholds corrected for multiple testing using a weighted Bonferroni correction that controls the family-wise error rate (FWER). Based on variant annotation classes the weights used are the predicted functional impact of the class. A total of 45,078,764 sequence variants were tested in either deCODE or the UK Biobank data. The significance thresholds are 1.8×10^{-7} for variants with high impact ($N = 12,456$), 3.5×10^{-8} for variants with moderate impact ($N = 235,454$), 3.2×10^{-9} for low-impact variants ($N = 3,334,594$), 1.6×10^{-9} for other variants in Dnase I hypersensitivity sites ($N = 5,928,505$) and 5.3×10^{-10} for all other variants ($N = 35,567,755$).”

References

1. Loh, P.-R. *et al.* Efficient Bayesian mixed-model analysis increases association power in large cohorts. *Nat. Genet.* **47**, 284–290 (2015).
2. Astle, W. J. *et al.* The Allelic Landscape of Human Blood Cell Trait Variation and Links to Common Complex Disease. *Cell* **167**, 1415–1429.e19 (2016).
3. Helgason, A., Hrafnkelsson, B., Gulcher, J. R., Ward, R. & Stefánsson, K. A populationwide coalescent analysis of Icelandic matrilineal and patrilineal genealogies: evidence for a faster evolutionary rate of mtDNA lineages than Y chromosomes. *Am. J. Hum. Genet.* **72**, 1370–1388 (2003).
4. Jónsson, G., Magnússon, M. S. & Hagstofa, Í. *Hagskinna: Icelandic historical statistics : sögulegar hagtölur um Ísland.* (1997).
5. Gudbjartsson, D. F. *et al.* Large-scale whole-genome sequencing of the Icelandic population. *Nat. Genet.* **47**, 435–444 (2015).
6. Lelliott, P. M., McMorran, B. J., Foote, S. J. & Burgio, G. The influence of host genetics on erythrocytes and malaria infection: is there therapeutic potential? *Malar. J.* **14**, 289 (2015).
7. Price, A. L. *et al.* The impact of divergence time on the nature of population structure: an example from Iceland. *PLoS Genet.* **5**, e1000505 (2009).
8. Sveinbjornsson, G. *et al.* HLA class II sequence variants influence tuberculosis risk in populations of European ancestry. *Nat. Genet.* **48**, 318–322 (2016).

REVIEWERS' COMMENTS:

Reviewer #1 (Remarks to the Author):

As you have written in the response, can you explicitly say in the manuscript "It is quite unusual to identify multiple independent variants in the same gene, especially when these variants are coding and have large opposing effects on a trait.", and then say that is why you focused on the ACO1 gene?

Need CIs or some assessment of error on heritability estimates in text.

Reviewer #2 (Remarks to the Author):

In this resubmission, the authors have greatly improved the narrative and readability. The findings are very interesting and important.

Minor points:

1. Second paragraph- English usage- should be large numbers of sequence variants have been associated with variations in hemoglobin conc...
2. A recent study revealed 140 sequence variants
3. page 5, please specify with INCREASED SERUM ferritin levels (top paragraph)
4. Should be [4Fe-4S] (lower case e)
5. bottom page 9, should be and is thus predicted..

Reviewer #3 (Remarks to the Author):

This reviewer could not assess the changes in the manuscript are they were not labelled. Based on the answer to reviewers' questions, the revised manuscript addressed some the issues raised but further clarifications are still needed. There is still lack of clarity on the focus of the paper. The introduction says "we focus on the rare missense and loss-of-function variants with large effects on hemoglobin concentration." but the paper just describes the ACO1 findings.

Answer to R#3, question 1b. for replication, please remove the overlap samples UK Biobank).

R#3, question 2. Answer to R#1 comment 1b does not include the steps leading to focus on the ACO1 gene. Perhaps a figure with the steps for the study design can clarify this given both R#1 and R#3 have the same questions. If variants at the ACO1 gene were the only significant findings of the study, then state this in the result section. Adding Manhattan and QQ plots for discovery would be helpful to clarify this. The study should also report lambdas for discovery studies.

R#3, question 3. Did you use conditional analysis for this, if so, describe in methods.

R#3, question 16a. I don't think adjusting for age at death is the appropriate adjustments to do. I noticed that the justification for this variable is not included in the revised manuscript. There are concerns that some of the measures were obtained during illness which can bias the results (anemia due to blood dilution, acute illnesses, bleeding). The authors need to either remove measures from hospitalizations (just use ambulatory measures) and/or remove measures close to time of death. This is relevant considering answer to question #17 that states that most of the measures are from hospitalizations.

Table 1. include the minor allele count for low frequency variants.

Results, lines 84-85, sentence "Subsequently, we tested the 34 remaining coding variants in ACO1 for association with hemoglobin concentration and found three additional associations after accounting for multiple testing". The rationale for the strategy including the selection of the variants and choice of

significance thresholds needs further support. This is not described in methods.

For Icelandic study, please state if informed consent was obtained.

Point-by-point answer to reviewers - second review

We would like to thank the three reviewers for their critique in the second review. We have made substantial changes to the manuscript in accordance to their comments and believe that we have addressed all their concerns.

All changes to the manuscript from previous review can be reviewed in track-changes. Line numbers below refer to edits in the manuscript without track changes..

Reviewers' comments:

Reviewer #1 (Remarks to the Author):

Comment #1

As you have written in the response, can you explicitly say in the manuscript "It is quite unusual to identify multiple independent variants in the same gene, especially when these variants are coding and have large opposing effects on a trait.", and then say that is why you focused on the ACO1 gene?

Response to Comment #1

To make this message clearer for the reader we have now emphasized that it is unusual to identify multiple independent variants in the same gene and give a rationale for the focus on *ACO1*.

Page 3, lines 70-79 in the Introduction section now reads (changes underlined):

"It is unusual to observe variants in the same gene that associate with a phenotype independently of each other. This is especially true when the observed variants are rare, coding and have large opposing effects on a trait. Therefore, of the loci harboring common and rare variants associated with hemoglobin concentration, we chose to focus on the ACO1 locus to better understand the effects of sequence variation in this gene on erythropoiesis in humans. ACO1 is of particular interest as this is a well characterized gene in cell and animal models, but little has been reported on the effects of sequence variation on this gene in humans. We report eight variants associated with hemoglobin concentration in ACO1, encoding cytosolic aconitase 1 (aka Iron-responsive element binding protein 1), a protein involved in cellular iron-sensing. These include six rare coding variants, where four associate with increased and two with decreased hemoglobin concentration. "

Comment #2

Need CIs or some assessment of error on heritability estimates in text.

Response to Comment #2

Confidence intervals on heritability estimates as an assessment of error have been added to the text.

Page 4, lines 92-94, now read (changes underlined):

“Heritability of hemoglobin concentration in the Icelandic population was estimated to be 0.20 (95% CI 0.19-0.21) and 0.29 (95% CI 0.29-0.30) using parent-offspring and sibling correlations, respectively.”

Reviewer #2 (Remarks to the Author):

In this resubmission, the authors have greatly improved the narrative and readability. The findings are very interesting and important.

Minor points:

Comment #1

1. Second paragraph- English usage- should be large numbers of sequence variants have been associated with variations in hemoglobin conc...

Response to Comment #1

We have made changes to the sentence in accordance with the reviewer's suggestion.

Page 3, lines 56-57 now read (changes underlined):

“A large number of sequence variants have been associated with variation in hemoglobin concentration through genome-wide association studies (GWAS).”

Comment #2

2. A recent study revealed 140 sequence variants

Response to Comment #2

Page 3, lines 57-59 now read (changes underlined):

“In particular, a recent study using a combined cohort of the UK Biobank and Interval studies revealed 140 sequence variants associated with hemoglobin concentration.”

Comment #3

3. page 5, please specify with INCREASED SERUM ferritin levels (top paragraph)

Response to Comment #3

We have made changes to the first paragraph on page five as suggested.

Page 5, line 129-131 now reads (changes underlined):

“In Iceland, we detect an association of one of the variants, Cys506Ser, with increased serum ferritin levels but none of the other variants are significant after accounting for multiple testing.”

Comment #4

4. Should be [4Fe-4S] (lower case e)

Response to Comment #4

We thank the reviewer for pointing this out. We have now made sure all [4Fe-4S] have lower case e.

Comment #5

5. bottom page 9, should be and is thus predicted.

Response to Comment #5

We have made changes to the sentence in accordance to your suggestion.

Page 10, lines 305-307 now reads (changes underlined):

“The variant is within domain 2 of the protein and is thus predicted to truncate the protein at amino acid 334 out of the 889 amino acid of the full-length protein.”

Reviewer #3 (Remarks to the Author):

This reviewer could not assess the changes in the manuscript as they were not labelled. Based on the answer to reviewers' questions, the revised manuscript addressed some of the issues raised but further clarifications are still needed. There is still lack of clarity on the focus of the paper.

Comment #1

The introduction says “we focus on the rare missense and loss-of-function variants with large effects on hemoglobin concentration.” but the paper just describes the ACO1 findings.

Response to Comment #1

Regarding other findings of the hemoglobin GWAS meta-analysis in Iceland and UK, in the main text we now mention the total number of loci with genome-wide significant signals, the replication of known hits, and the number of loci with genome wide significant rare coding variants. Subsequently, we underline the uniqueness of the finding at the ACO1 locus. We also provide supplementary data summarising these findings (Supplementary Data 1, Supplementary Data 2 and Supplementary Data 3). Also, summary statistics are provided for the GWAS meta-analysis of hemoglobin levels for all tested variants in Iceland and the UK datasets (Data availability).

Page 4, lines 95-109 in the Results section now read (changes underlined):

"We observe 334 loci harboring sequence variants reaching genome-wide significance (Supplementary Fig. 3 and Supplementary Data 1). We provide summary statistics for the GWAS meta-analysis of hemoglobin concentration in Iceland and the UK for all tested variants (Supplementary Data 1 and Data Availability section). In total, 138 variants at 121 loci have previously been reported to associate with hemoglobin levels in populations of European descent, for which, we provide robust replication (98%) and demonstrate consistency of effect in the Icelandic and UK datasets in the current study (Supplementary Data 2). We observe genome-wide significant associations of 22 rare coding variants (MAF < 1 %) were observed at 13 out of the 334 loci

associated with hemoglobin level (Supplementary Fig. 3 and Supplementary Data 3). We observe independent rare coding variants with opposing effects at both the EGLN2 and ACO1 loci. Rare coding variants in EGLN2 were reported by Astle et al., whereas none have been reported in ACO1.

Five variants in ACO1, encoding cytosolic aconitase 1, also known as Iron-responsive element binding protein 1 (IRP1), associate genome-wide significantly with hemoglobin concentrations, of which three are coding and rare, and one common non-coding variant rs7045087 represents a previously reported intergenic association (Table 1)."

Comment #2

Answer to R#3, question 1b. for replication, please remove the overlap samples UK Biobank).

Response to Comment #2

We do not have information on which individuals made up the fraction of the UK biobank dataset used in the publication by Astle et al. We are only able to infer the size of the overlap between the UK Biobank datasets based on the reported sample size in the study by Astle et al. That study included 87,000 UK biobank participants¹. Thus, at least 78% of the 398,000 participants we included from the current UK biobank dataset were not a part of the study by Astle et al. and therefore the overlap is at most 22 %.

In the current study we assessed sequence variants reported by Astle et al. to associate with hemoglobin among 173,480 European-ancestry participants which include 87,000 individuals from the UK biobank within a larger set of 398,000 individuals from the UK biobank . For 86% of the variants the association is more significant in the larger UK dataset (n = 104 out of 121 tested) (Supplementary Data 2). This demonstrates that the initial findings are not driven by the smaller UK biobank dataset in the initial study by Astle et al.

Also, the effects of the reported hemoglobin associated variants in the current meta-analysis are highly consistent between the Icelandic and UK Biobank datasets (see Supplementary Data 2).

To demonstrate the robust replication of previously reported hemoglobin associated variants in our GWAS meta-analysis of hemoglobin levels, we have added heterogeneity P-values to Supplementary Data 2 and we made changes to the “Reported hemoglobin associated variants” subsection in the Results to emphasize this point.

Page 9, lines 245-254, now read (changes underlined):

“We show association results for 175 reported associations of sequence variants with hemoglobin concentration, 138 of which have previously been reported in populations of European descent. The large majority of reported variants (N = 119) come from the hitherto largest hemoglobin GWAS reported by Astle et al. in 2016, where the UK biobank participated with 87 K individuals¹, which comprises 22% of the UK biobank dataset used in the current study (Supplementary data 2)

Out of the 138 variants reported in European populations, 131 were tested in both the Icelandic and UK datasets and all show a direction of effect that is consistent with the initial report. In Iceland, 113 out of the 131 variants replicate (Supplementary data 2 and Supplementary Fig. 10). For the combined Icelandic and UK datasets 129 out of 131 variants replicate.”

Comment #3a

R#3, question 2. Answer to R#1 comment 1b does not include the steps leading to focus on the ACO1 gene. Perhaps a figure with the steps for the study design can clarify this given both R#1 and R#3 have the same questions.

Response to Comment #3a

We thank the reviewer for pointing this out. To make these messages clearer for the reader we now include a flowchart of the study design that describes the steps leading to the focus on the ACO1 locus (Supplementary Figure 1).

See also the response to comment #1 from Reviewer#1

Supplementary Figure 1. A flowchart describing the study design of the hemoglobin concentration meta-analysis of Icelandic and UK Biobank datasets, and a results summary.

Comment #3b

R#3, question 2. If variants at the ACO1 gene were the only significant findings of the study, then state this in the result section. Adding Manhattan and QQ plots for discovery would be helpful to clarify this.

Response to Comment #3b

A statement has now been added to the Result section regarding additional findings of the current hemoglobin GWAS meta-analysis. We also provide a manhattan plot for the meta-analysis (Supplementary Figure 3), and QQ-plots of the GWAS results in Iceland and the UK (Supplementary Figure 11). Also, summary statistics are provided for the GWAS meta-analysis of hemoglobin levels for all tested variants in Iceland and the UK datasets.

See also the response to comment #1 from the same reviewer

Supplementary Figure 11 Quantile–quantile plot (QQ–plot) for all sequence variants for the Icelandic and UK datasets included in the meta–analysis of hemoglobin concentration meta–analysis. The estimated correction factor for Hemoglobin concentration based on LD score regression was 0.68 for the additive model in the Icelandic sample and 1.40 in the UK Biobank. The Y axis shows observed $-\log_{10}$ P–values, and the X axis shows the expected $-\log_{10}$ P–values.

Supplementary Figure 2 Manhattan plot for hemoglobin concentration meta-analysis association results using a combined set of 684,122 individuals from Iceland and the UK. Variants are plotted by chromosomal position (x-axis) and $-\log_{10}(P)$ (y-axis). Blue=loci harboring previously reported hemoglobin concentration associated variants, green=Novel loci harboring genome-wide significant hemoglobin concentration associated variants. The thirteen genes that harbor rare (MAF < 1%) coding sequence variants are labeled in red.

Comment #3c

The study should also report lambdas for discovery studies.

Response to Comment #3c

In our response to comment #4 from reviewer #1 in our previous response we removed the following sentence from the Methods section:

“The estimated correction factor for Hemoglobin concentration based on LD score regression was 0.68 for the additive model in the Icelandic sample and 1.40 in the UK Biobank.”

We have reintroduced this sentence along with the explanation given in the previous response. Also, we have included a QQ-plot before and after correction for both the Icelandic and UK datasets.

Pages 15-16, lines 437-448, now read (changes underlined):

“The estimated correction factor for hemoglobin concentration based on LD score regression was 0.68 for the additive model in the Icelandic sample and 1.40 in the UK Biobank. In Iceland, when testing the association of sequence variants with quantitative traits, a BOLT linear mixed model was applied. These models are now widely used as they account for cryptic relatedness while also increasing power². One-step in the BOLT-LMM procedure (step 1b) is to calibrate the χ^2 test statistic by calculating a constant calibration factor. To compute the calibration constant BOLT-LMM rapidly computes the prospective statistic at 30 random SNPs by applying conjugate gradient iteration. However, this scaling was not applied to the test statistic in the Icelandic association model. Therefore, when we applied the LD score regression and estimate a correction factor from the regressions intercept it was shifted by this constant factor. The correction factor can thus indeed be below one due to the calibration factor (Supplementary Fig. 11). The intercept is therefore not comparable to correction factors obtained from standard genomic control methods, and should not be interpreted as such.”

Comment #4

R#3, question 3. Did you use conditional analysis for this, if so, describe in methods.

Response to Comment #4

Conditional analysis was performed for the *ACOI* variants reported in the current study.

Results from the conditional analysis are summarized in supplementary table 3 which is referred to in the end of third paragraph of the results (page 5, line 118) and in the common variant subsection of the results (page 8, line 232).

We have added a description of the conditional analysis performed to the Online Methods section in the subsection called “Association analysis”.

Page 16, lines 449-450, now read (changes underlined):

“Expected allele counts for sequence variants were used as covariates in the regression to test for association with other sequence variants conditional on their effects.”

Comment #5a

R#3, question 16a. I don't think adjusting for age at death is the appropriate adjustments to do. I noticed that the justification for this variable is not included in the revised manuscript.

Response to Comment #5a

Age at death is not used as a covariate in the linear mixed model when testing for the association of sequence variants with quantitative traits. For binary phenotypes, current age is used as a covariate for living individuals in the logistic regression model, and alternatively age at death is used for the deceased.

To make this clear to the reader we have made changes to the subsection "Association models" in the "Online methods" section of the manuscript

Page 15, lines 420-428, now read (changes underlined):

"We performed a meta-analysis on 286,622 individuals from Iceland and 397,500 individuals from the UK Biobank with at least one hemoglobin concentration measurement. In Iceland, quantitative traits were tested using a linear mixed model implemented in BOLT-LMM². We tested 37,592,353 variants (with imputation information > 0.8 and MAF > 0.01%) identified from the whole-genome sequencing of 28,075 Icelanders (~9% of the population) for association with hemoglobin concentration. For binary phenotypes, sex, county of birth, current age or age at death (first and second order terms included), blood sample availability for the individual and an indicator function for the overlap of the lifetime of the individual with the time span of phenotype collection were included as covariates in the logistic regression model."

Comment #5b

There are concerns that some of the measures were obtained during illness which can bias the results (anemia due to blood dilution, acute illnesses, bleeding). The authors need to either remove measures from hospitalizations (just use ambulatory measures) and/or remove measures close to time of death. This is relevant considering answer to question #17 that states that most of the measures are from hospitalizations.

Response to Comment #5b

The studied populations in Iceland and UK are quite different, however when testing reported variants we observe similar effect sizes.

In Iceland, the median age of measurement is 72 years of age, whereas it is 60 in the UK. Participants had on average over 6 measurements in Iceland (geometric mean = 6.4), we used the mean value for those with multiple measurements in the analysis. Also, we used the raw hemoglobin concentration measurements to derive anemia and polycythemia status when measurements were always above or below their respective diagnostic thresholds. In the UK, a single measurement was performed on blood samples obtained from UK Biobank assessment centre visit.

In Iceland, the large majority (~88 %) of hemoglobin measurements are made by the hospital laboratory which also measures samples from primary care clinics in the capital area, and is not necessarily as a result of hospitalization. In contrast, in the UK, a single hemoglobin measurement is done in a research context at a UK Biobank assessment center.

Hemoglobin measurements are available for a large fraction of Icelanders (287 K individuals out of a total population size of 360 K ~ 80 %). For the UK dataset, from approximately 9.2 million individuals invited to join the UK Biobank study ~5.5 % (~505 K individuals) were recruited and provided full informed consent to participate in UK Biobank study and completed a health assessment that includes a hemoglobin concentration measurement. A recent study by Fry et al. has concluded that the UK Biobank is not representative of the sampling population and that there is evidence of a “healthy volunteer” selection bias³.

Thus, the differences in the raw hemoglobin measurements between the two populations are a function of the differences in recruitment practices mentioned above (Supplementary Table 1 and Supplementary Figure 2).

The main aim of this paper is discovering associations of rare coding sequence variants with a large effect on hemoglobin concentration. To demonstrate robustness, in Icelandic and UK datasets we replicate the large majority (129 of 131) of hemoglobin associated variants reported in European

populations. Also, we compared effects in standardized and raw scale (g/L) for the 131 hemoglobin associated variants reported in European populations to explore whether there is a difference in effect estimates between the Icelandic and UK datasets (Supplementary Fig. 10). There are 27% higher effect estimates on the standardized scale in the UK dataset than in the Icelandic one (ratio of effect UK/Iceland = 1.27 (95% CI 1.23-1.32)). We note that the variance of raw hemoglobin concentration is higher in the Icelandic dataset than the UK one (standard deviation of raw hemoglobin concentration: Iceland = 15.5 g/L, UK = 12.2 g/L) (Supplementary Table 1). Once effect estimates are converted to raw scale (g/L) the effects are almost identical in the Icelandic and UK datasets (ratio of effect UK/Iceland = 1.02 (95% CI 0.99-1.06)) (Supplementary Fig. 10). Thus, it appears that the difference in effect estimates on the standardized scale between UK and Iceland can largely be explained by the higher variance in hemoglobin concentration in Iceland.

For variants at the ACO1 locus, no significant difference between the effects in Iceland and UK on hemoglobin levels is observed (heterogeneity P-value < 0.05/4 = 0.0125) (Table 1). In addition, when testing the variants Cys506Ser and Lys334Ter at *ACO1*, which are only present in Iceland, we presented the raw values of carriers and non-carriers stratified by age and sex and observe no indication that the effects are limited to sex or a specific age group (Supplementary Figure 7).

The UK and Iceland datasets included in the present analysis are diverse in regard to recruitment practices, but we still observe similar effects of sequence variants in the datasets. Despite the fact that the distribution of raw hemoglobin values is different in Iceland and the UK, the effect of the reported variants is of similar size in standardized values.

To emphasize the robust replication of previously reported hemoglobin associated variants and consistency of effect between the Icelandic and UK datasets in the current study we have made changes to the “Reported hemoglobin associated variants” subsection in the Results. Also, we have added a scatter plot comparing the effect of reported hemoglobin associated variants between the Iceland and UK datasets in standardized and raw values (Supplementary Fig. 10).

Page 9, lines 245-272, now read (changes underlined):

“We show association results for 175 reported associations of sequence variants with hemoglobin concentration, 138 of which have previously been reported in populations of European descent. The large majority of reported variants (N = 119) come from the hitherto largest hemoglobin GWAS reported by Astle et al. in 2016, where the UK biobank participated with 87 K individuals¹, which comprises 22% of the UK biobank dataset used in the current study (Supplementary data 2)

Out of the 138 variants reported in European populations, 131 were tested in both the Icelandic and UK datasets and all show a direction of effect that is consistent with the initial report. In Iceland, 113 out of the 131 variants replicate (Supplementary data 2

and Supplementary Fig. 10). For the combined Icelandic and UK datasets 129 out of 131 variants replicate. We also compared effects in standardized and raw scale (g/L) for the 131 hemoglobin associated variants reported in European populations to explore whether there is a difference in effect estimates between the Icelandic and UK datasets (Supplementary data 2 and Supplementary Fig. 10). There are 27% higher effect estimates on the standardized scale in the UK dataset than in the Icelandic one (ratio of effect UK/Iceland = 1.27 (95% CI 1.23-1.32)). We note that the variance of raw hemoglobin concentration is higher in the Icelandic dataset than in the UK one (standard deviation of raw hemoglobin concentration: Iceland = 15.5 g/L, UK = 12.2 g/L) (Supplementary Table 1). Once effect estimates are converted to raw scale (g/L) the effects are almost identical in the Icelandic and UK datasets (ratio of effect UK/Iceland = 1.02 (95% CI 0.99-1.06)) (Supplementary Fig. 10). Thus, it appears that the difference in effect estimates on the standardized scale between UK and Iceland can largely be explained by the higher variance in hemoglobin concentration in Iceland.

The UK and Iceland datasets included in the present analysis are diverse in regard to recruitment practices^{9,29}. Despite differences in age, population coverage, number, and purpose of measurements between the Icelandic and UK datasets, that are reflected in differences in the distribution of raw hemoglobin values (Supplementary Table 1 and Supplementary Fig. 2), we still observe similar effect of sequence variants on hemoglobin concentration in the two datasets (Table 1, Supplementary Data 2, and Supplementary Fig. 10).”

Supplementary Fig. 10. A scatter plot showing hemoglobin concentrations effects for 131 variants reported to associate with hemoglobin concentration in Europeans for the Icelandic (x-axis) vs. UK datasets (y-axis). The blue line shows inverse-variance weighted (IVW) slope
 A) Hemoglobin concentration effects are shown in standardized values. Ratio of effect UK/Iceland = 1.27 (95% CI 1.23-1.32)
 B) Hemoglobin concentration effects are shown in g/L. Ratio of effect UK/Iceland = 1.02 (95% CI 0.99-1.06).

Comment #6

Table 1. include the minor allele count for low frequency variants.

Response to Comment #6

Minor allele count has been added to Table 1.

Comment #7

Results, lines 84-85, sentence “Subsequently, we tested the 34 remaining coding variants in ACO1 for association with hemoglobin concentration and found three additional associations after accounting for multiple testing”. The rationale for the strategy including the selection of the variants and choice of significance thresholds needs further support. This is not described in methods.

Response to Comment #7

We have added a description of the gene-based strategy to the Online Methods section.

Page 16, lines 459-462, now read (changes underlined):

“Given that we observe genome-wide significant associations to hemoglobin levels corresponding to coding variants in ACO1, we decided to test all ACO1 coding variants with hemoglobin levels. In total, we tested 34 coding variants in ACO1 and apply a Bonferroni correction significance threshold of $0.05/34=1.5 \times 10^{-3}$ ”.

Comment #8

For Icelandic study, please state if informed consent was obtained.

Response to Comment #8

All participating individuals, or their guardians, who donated blood provided written informed consent and approval for the study was provided by the Icelandic National Bioethics Committee (ref: VSNb2015010033-03.12). This is stated in the second paragraph of the “Study Subjects” subsection in the Online Methods. However, it is not explicitly stated that this refers to participating Icelandic individuals.

To make this message clearer for the reader we have changed the second paragraph of the “Study Subjects” subsection in the Online Methods.

Page 13, lines 356-357, now read (changes underlined):

“All participating Icelandic individuals who donated blood, or their guardians, provided written informed consent.”

1. Astle, W. J. *et al.* The Allelic Landscape of Human Blood Cell Trait Variation and Links to Common Complex Disease. *Cell* **167**, 1415–1429.e19 (2016).
2. Loh, P.-R. *et al.* Efficient Bayesian mixed-model analysis increases association power in large cohorts. *Nat. Genet.* **47**, 284–290 (2015).
3. Fry, A. *et al.* Comparison of Sociodemographic and Health-Related Characteristics of UK Biobank Participants With Those of the General Population. *Am. J. Epidemiol.* **186**, 1026–1034 (2017).